

# Exploring aerosol-cloud interactions over eastern China and its adjacent ocean using the WRF-SBM-MOSAIC model

Jianqi Zhao[1], Xiaoyan Ma[1], Johannes Quaas[2], Hailing Jia[2]

[1]Key Laboratory for Aerosol-Cloud-Precipitation of China Meteorological Administration, Nanjing University of Information Science & Technology, Nanjing 210044, China
[2]Institute for Meteorology, Leipzig University, Leipzig, Germany

*Correspondence to*: Xiaoyan Ma (xma@nuist.edu.cn)

**Abstract.** This study aims to explore aerosol-cloud interaction over eastern China (EC) and its adjacent ocean (ECO) in boreal winter by coupling of a spectral-bin cloud microphysics (SBM) and an online aerosol module (MOSAIC) in WRF-Chem, with the support of four-dimensional data assimilation. The evaluation shows that assimilation has an overall positive impact on the simulation, and the coupling system reproduces the satellite-retrieved cloud parameters while exhibiting significantly improved simulation ability compared to the original SBM scheme as well as the bulk microphysical and MOSAIC coupling system. Differences in aerosol composition and physical processes lead to clear discrepancies in the aerosol-cloud interactions of EC and ECO during the simulation period. In EC with the gradual increase of aerosol number concentration ($N_{aero}$), cloud droplet number concentration ($N_d$) first increases then decreases and fluctuates around 800 cm$^{-3}$, while $N_d$ in ECO increases faster initially, but soon its activation is suppressed by aerosol hygroscopicity and high activation threshold of numerous small particles, and almost no additional cloud droplets are produced. In terms of rapid adjustments, more bursty atmospheric supersaturation and lack of subsequent water cause cloud liquid water content (CLWC) in EC to increase explosively with $N_d$ when there are few cloud droplets, but only maintains a low increase rate with further increasing $N_d$. ECO exhibits a fast increase in CLWC with $N_d$ at high proportion of naturally emitted large aerosol particles, but its CLWC increase gradually stagnates as $N_d$ increases. For non-precipitating clouds with less water content, CLWC in EC increases slowly with $N_d$, but can maintain a stable trend. While ECO, which relies mainly on large scale water and temperature variations to reach supersaturation, the increase in $N_d$ leads to a decrease in CLWC.

## 1 Introduction

Atmospheric aerosols have significant effects on the Earth's radiation balance, water cycle, and climate system through direct absorption and scattering of solar radiation as well as indirect effects on cloud formation and development by acting as cloud condensation nuclei (CCN) (Carslaw et al., 2010; Wilcox et al., 2013; Tian et al., 2021). The latter, known as the aerosol indirect effect, or more recently by the Intergovernmental Panel on Climate Change (2013) defined as effective radiative forcing due to aerosol-cloud interactions, $RF_{aci}$, remain a challenging scientific topic in climate assessment and prediction because of its complex mechanisms and high uncertainties (Church et al., 2013; Jia et al., 2019a; Arias et al., 2021).



Twomey (1977) pointed out that under a constant cloud water content, the activation of atmospheric aerosol particles entering into clouds leads to an increase in cloud droplet number concentration ($N_d$), and, even if this implies a decrease in droplet size, to an increase in cloud albedo. This mechanism, termed the aerosol first indirect effect, is revealed to be the key driver of aerosol indirect effect, besides, the rapid adjustments also contribute significantly (Quaas et al., 2020). Two key

competing mechanisms exist in the latter, one of which is that an increase in $N_d$ causes a decrease in precipitation efficiency and with this, a co-increase in cloud liquid water path (CLWP) and cloud fraction (CF), this mechanism dominates in precipitation clouds (Albrecht, 1989). The other mechanism dominates in non-precipitating clouds, i.e., with limited water content, the decrease in droplet size reduces sedimentation velocity and increases cloud-top liquid water content, resulting in additional cloud top cooling and pushing further entrainment and evaporation (Bretherton et al., 2007). Moreover, as cloud

droplets decrease in size, their ratio of surface area to volume is higher and evaporation is faster, resulting in further enhancement of the negative buoyancy at cloud top (Small et al., 2009). Numerous researches have been conducted to assess the contribution of these three mechanisms. Statistical analysis based on satellite-retrieved data indicates that the CLWP of marine low clouds exhibits a weak decreasing trend with rising $N_d$ caused by aerosol increase (Michibata et al., 2016; Rosenfeld et al., 2019). Gryspeerdt et al. (2019) found that CLWP is positively correlated with $N_d$ at low $N_d$ and at droplet size greater

than the precipitation threshold, i.e., delayed precipitation leads to increased CLWP. In contrast, for cloud with high $N_d$ and low possibility of precipitation, CLWP shows a negative correlation with $N_d$. In this case, the increase of aerosol leads to the decrease of cloud droplet size and the increase of $N_d$, which in turn accelerates the mixing and evaporation process and makes CLWP decrease. The CLWP response to aerosols differs clearly between precipitation and non-precipitation clouds because of the significant influence of precipitation process on CLWP (Christensen and Stephens, 2012). CLWP has a significant

positive correlation with the aerosol index (AI) in precipitation clouds, and the opposite in non-precipitation clouds (Chen et al., 2014). Furthermore, the response of CLWP to aerosol highly depends meteorological conditions. Chen et al. (2014) indicated that CLWP and aerosol concentration show a negative correlation when entrainment mixing exerts a marked impact on the cloud-side evaporation process (which usually occurs under free troposphere with dry and unstable atmosphere), and this relationship shifts to positive as the atmosphere becomes moist and stable. Such statistical analysis, however, suffers

severely from retrieval uncertainties (Arola et al., 2022). In turn, also "opportunistic experiments" such as the analysis of ship and pollution tracks hint at a decrease in CLWP but an increase in cloud horizontal extent in response to aerosol increases (Toll et al., 2019; Christensen et al., 2022). In spite of considerable efforts in recent researches to unravel aerosol-cloud interactions, it remains challenging to distinguish and quantify underlying mechanisms of aerosol-cloud interactions under diverse air pollution and meteorological conditions.

In order to further resolve the mechanisms of aerosol-cloud interactions, the proper use of numerical simulations is necessary. However, current global climate models (GCMs) have difficulties in accurately representing the response of cloud to aerosol, which is mainly due to (1) the limitation of coarse model resolution, (2) the absence of sufficient consideration of cloud droplet spectral characteristics, and (3) the fact that most current GCMs parameterize the precipitation mechanism acting through the autoconversion process as an inverse function of $N_d$, without accurate representation of entrainment-mixing



processes (Quaas et al., 2009; Bangert et al., 2011; Michibata et al., 2016; Zhou and Penner, 2017). Regional climate models (RCMs) with higher resolution and finer physical parameterization can effectively compensate for at least some of these shortcomings and better reproduce the physical processes, which help to further distinguish and quantify the aerosol-cloud interaction mechanisms (Li et al., 2008; Bao et al., 2015). The Weather Research and Forecasting model (WRF) has been widely used in regional numerical simulation research because of its advanced technology in numerical calculation, model

framework, and program optimization, which has many advantages in portability, maintenance, expandability, and efficiency (Maussion et al., 2011; Islam et al., 2015; Xu et al., 2021). The chemistry-coupled version of the WRF model (WRF-Chem) allows to simulate the spatial and temporal distributions of reactive gases and aerosol, spatial transport and their interconversion while simulating meteorological fields and atmospheric physical processes, which helps to reproduce the aerosol-cloud interaction scenarios in real cases (Tuccella et al., 2012; Sicard et al., 2021). Bulk and bin approaches are

commonly utilized to simulate regional cloud microphysical processes. Bulk schemes diagnose the size distribution of hydrometeor based on different predicted bulk mass (one-moment schemes) or number and mass mixing ratios (double-moment schemes) and assumed size distribution shapes, showing significant limitations in reproducing processes such as condensation, deposition and evaporation (Lebo et al., 2012; Wang et al., 2013; Fan et al., 2015). Bin schemes predict the size distribution of hydrometeors based on a number of discrete bins, enabling better representation of cloud microphysical

processes. As stated by Khain et al. (2015), numerous works have demonstrated that bin schemes outperform bulk schemes in simulations, but for reasons such as computational cost, WRF-Chem currently only provides the coupling of bulk microphysical schemes with an online aerosol module (Gao et al., 2016). Bin microphysical schemes with high computational costs are usually running with predefined, composition-fixed aerosol spectra and simple aerosol budget treatment, lacking adequate consideration of aerosol sources and sinks, making it difficult to reproduce real-time aerosol information, which

greatly hinders the investigation of aerosol-cloud interactions (Khain et al., 2009; Qu et al., 2017; Gibbons et al., 2018; Chen et al., 2019). Benefiting from advances of computational science, the coupling of bin microphysics and online aerosol, which was prohibited previously due to high computational costs, is starting to become feasible (Gao et al., 2016). By coupling them, more reliable aerosol information is provided for cloud microphysical simulations, and more accurate microphysical parameters are also offered to aerosol-chemistry simulations, which are of great help to reproduce real conditions as well as to

distinguish and quantify aerosol-cloud interaction mechanisms.

       Eastern China (EC) is one of the most human-active regions worldwide, resulting in numerous anthropogenic aerosol emissions. The contrast between the high aerosol-content air masses of EC and the relatively clean air masses of the Pacific Ocean makes EC and its adjacent ocean (ECO) ideal regions for exploring aerosol-cloud interactions (Fan et al., 2012; Wang et al., 2015; Zhang et al., 2021). It is shown that low clouds contribute the most to the Earth's energy balance due to their broad

coverage and the albedo effect governing their impact on emitted thermal radiation (Hartmann et al., 1992). The statistics of Niu et al. (2022) for the satellite data from 2007-2016 show that low clouds in EC and ECO occur most frequently in winter, reaching more than 50%, with stratocumulus clouds, which are sensitive to aerosol variations (Jia et al., 2019b), constituting more than 70% of the low clouds. Therefore, the EC and ECO aerosol-cloud response in winter is an ideal entry point to



investigate aerosol indirect effects. Based on the coupling of the spectral-bin microphysics (SBM) scheme and the Model for

Simulating Aerosol Interactions and Chemistry (MOSAIC) in WRF-Chem (Khain et al., 2004; Fast et al., 2006), we investigate the aerosol-cloud interaction mechanisms of EC and ECO in winter by obtaining detailed and high-resolution aerosol, cloud parameters as well as meteorological information through reproduction of real scenarios.

The paper is structured as follows: Section 2 introduces the coupling of SBM with MOSAIC, model configuration, and observational data in the study, Section 3 presents the evaluation of simulated results and the analysis of aerosol-cloud

responses presented in the simulations, and the summary is given in Section 4.

## 2 Methods and Data

### 2.1 SBM and MOSAIC coupling system

This study is based on WRF-Chem v3.9, the full version of the SBM scheme coupled with the aerosol module (Khain et al., 2009). The scheme solves a system of prognostic equations for seven hydrometeor types (liquid drops, plate, columnar, and

branch ice crystal types, snow/aggregates, graupel, and hail/frozen drops) and CCN size distribution functions. Each size distribution function is structured by 33 mass doubling bins (i.e., the mass of the particle in the $k^{th}$ bin is twice that of the k-1$^{th}$ bin). The cloud microphysical processes described in the SBM contain nucleation of droplets and ice particles, freezing, melting, diffusion growth/evaporation of liquid drops, deposition/sublimation of ice particles, drop and ice collisions (Khain et al., 2004). In the current SBM scheme of WRF, the CCN distribution is prescribed and simply treated. This approach is

unable to consider realistic, temporally and spatially varying aerosol distribution that are consistent with the cloud and precipitation fields, and thus hinders resolving important aspects of aerosol-cloud interactions, which can be addressed effectively by coupling an online aerosol module.

The 4-bin version of the MOSAIC aerosol module used in the coupling system treats mass and number of nine major aerosol species, including sulfate, nitrate, sodium, chloride, ammonium, black carbon, primary organics, other inorganics, and

liquid water (Zaveri et al., 2008). The diameters of the 4 size bins are in the ranges from 0.039-0.156, 0.156-0.624, 0.624-2.5 and 2.5-10.0 μm, respectively, and aerosol particles are assumed to be internally mixed. The chemical composition within each bin is assumed identical. This scheme is capable of treating processes such as emissions, new particle formation, particle growth/shrinkage due to uptake/loss of trace gases, coagulation, dry and wet deposition (Zaveri et al., 2008). Selecting versions of MOSAIC that incorporate gas and liquid phase chemistry (currently coupled only to bulk cloud microphysical schemes)

allows further treatment of secondary organic aerosol generation, aerosol activation/resuspension, wet scavenging, and liquid phase chemistry within cloud (Sha et al., 2019; 2022). These versions of MOSAIC explicitly treat both unactivated (interstitial) and activated (cloud-borne) aerosols, which share the same particle size range, but the latter size refers to the dry diameter of the aerosol material within the cloud droplet. Replacing the bulk scheme with the SBM to couple with MOSAIC, as MOSAIC provides the SBM with more detailed aerosol information, SBM also offers MOSAIC meteorological field and cloud

microphysical information interactively, thus greatly promotes the model's ability to reproduce aerosol-cloud interactions.





The structure of the coupling system is presented in Fig. 1. The aerosol number concentration, mass and composition information provided by MOSAIC are treated in SBM with aerosol-cloud parameterization, and in turn, the updated aerosol information and cloud microphysical parameters required to calculate the processes such as gas in-cloud removal as well as aerosol and gas below-cloud removal are fed back to MOSAIC. For MOSAIC, we disabled the calculation of in-cloud aerosol removal, aerosol activation and resuspension processes. For SBM, the major modifications include (1) adjusting the SBM CCN bins to fit the MOSAIC aerosol size ranges, (2) allocating the 4-bin MOSIAC aerosols to the 33-bin CCN in SBM; (3) modifying the SBM aerosol activation parameterization to fit the weak updrafts in winter, and (4) calculating the in-cloud removal and resuspension of aerosols.

### SBM

1. Adjust SBM CCN bins based on MOSAIC aerosol
2. Allocate 4-sectional MOSAIC aerosol to SBM 33 CCN
3. Adapt SBM aerosol activation parameterization to small updrafts in winter
4. Calculate aerosol in-cloud removal and resuspension

Provide aerosol number concentration, mass and composition resolved in four bins

Update aerosol information and provide microphysical parameters needed by MOSAIC aerosol module

### MOSAIC aerosol module

1. Disable aerosol in-cloud wet removal
2. Disable aerosol activation and resuspension

**Figure 1.** Structure of SBM-MOSAIC coupling system

For the transfer of MOSAIC aerosol information to SBM, we first change the CCN radius range of the SBM from 1.23 nm-2 μm to 2.44 nm-5 μm, which covers the aerosol radius range of MOSAIC (19.5 nm-5 μm). Assuming that each bin of MOSAIC aerosol follows a lognormal distribution, the proportion of the $m^{th}$ MOSAIC aerosol bin assigned to the $n^{th}$ SBM CCN bin is given by:

$$P_{al}(m,n) = c_r \cdot \frac{1}{2} \cdot \left[ erfc\left( \frac{lnr_{low}(n) - lnr_g(m)}{\sqrt{2}ln\sigma_g} \right) - erfc\left( \frac{lnr_{up}(n) - lnr_g(m)}{\sqrt{2}ln\sigma_g} \right) \right] \quad (1)$$

where $r_{up}$ and $r_{low}$ are the upper and lower radius boundaries of the $n^{th}$ bin of SBM CCN, $r_g$ is the geometric mean radius of the $m^{th}$ bin MOSAIC aerosol, and $\sigma_g$ is the geometric standard deviation. Since 2 of the 3 log-normal distribution parameters are limited by the total aerosol number and dry-volume of a size bin, the value of $\sigma_g$ is set to 1.492 (Gao et al., 2016). Additionally, in order to prevent the MOSAIC aerosol from being assigned to an SBM CCN bin that exceeds its original size range under the assumption of lognormal distribution, the proportion of the excess is set to 0. A complementary factor $C_r$ (value of 1.09076) is set to make the sum of the corresponding 33 P_al for each MOSAIC bin to be 1.

For the calculation of the initial radius ($r_0$) of cloud droplets formed after aerosol activation, the original SBM scheme follows the parameterization of Ivanova (1977), i.e., when the aerosol dry radius ($r_d$) < 0.03 μm, $r_0$ is the equilibrium size corresponding to its critical supersaturation, and when $r_d$ > 0.03 μm, $r_0$ is 5 times of $r_d$. This parameterization is generally adopted for the simulation of deep convective clouds with strong updrafts and intense condensation. For better adaptation of



SBM to the simulation of weak updrafts in winter, we opt to switch to the parameterization of Kogan (1991) and Gao et al. (2016):

$$r_0 = \begin{cases} 5.8 \cdot w^{-0.12} \cdot r_d^{-0.214} \cdot r_d & r_d > r_d^*, w > 0 \\ 2.0 & r_d < r_d^*, w > 0 \\ 2.0 & w < 0 \end{cases} \tag{2}$$

where $r_0$ is in μm and $w$ is the updraft in m s$^{-1}$, $r_d^* = 0.09w^{-0.16}$ μm. The activated aerosol is subtracted from the interstitial aerosol as an item of aerosol in-cloud removal. For the characterization of cloud-borne aerosols, we use the same approach as for the MOSAIC and bulk cloud microphysical scheme coupling system (i.e., using the effective dry size of the aerosol material to represent the cloud-borne aerosols, and adopting the assumed/prescribed distributions to describe aerosol distribution among the different cloud droplet sizes instead of tracking them exactly). This approach is computationally less costly and requires relatively few changes to the original code.

For aerosol resuspension and in-cloud removal of cloud-borne aerosols (collisions that form rain, riming, and drop freezing), we refer to Gao et al. (2016), where the fractional losses of cloud-borne aerosol number and mass are calculated based on the changes in $N_d$ and droplet mass concentration before and after each process, the difference is that we trace these processes directly within the SBM rather than transferring the relevant parameters to an interface module. The below-cloud removal of aerosol as well as the in-cloud and below-cloud removals of gas are still treated by MOSAIC where the required parameters such as precipitation rates of hydrometeors and fractional cloud water sink are calculated in SBM.

## 2.2 Simulation Setup

The model domain is shown in Fig. 2, with the center point and the number of grid points as (32°N, 120°E) and 121×101, respectively. Since the main cloud types in winter are low clouds such as stratocumulus, which usually exceed 10 km in horizontal scale, the horizontal grid spacing is set to 15 km (treating sub-grid clouds with the Grell-3 scheme), and the integration step is set to 60 s. There are 48 vertical layers up to 50 hPa, with layer spacing extending from 40 m near the surface to 200 m at 3000 m altitude and over 1000 m above 10000 m altitude. The simulations run from 00:00:00 UTC on 1 Feb 2019 to 00:00:00 UTC on 13 Feb 2019, where the first 24 h are disregarded as spin-up and not involved in subsequent analyses. Meteorological initial and boundary conditions are obtained from the National Center for Environmental Prediction (NCEP) FNL global reanalysis data with 1° resolution and available every 6 h (https://rda.ucar.edu/datasets/ds083.2, last access: 19 March 2023), and anthropogenic emission sources come from the Multi-resolution Emission Inventory for China (MEIC) 2016 version developed by Tsinghua University (http://meicmodel.org.cn, last access: 19 March 2023). As presented in Fig. 2, the anthropogenic aerosols of EC (27.5°N-34°N, 112°E-120°E) and ECO (27.5°N-32.5°N, 123°E-128°E) are dominated by EC under winter monsoon, although the model domain contains countries and regions other than China, MEIC can satisfy the anthropogenic aerosol simulation of the region concerned in this study.





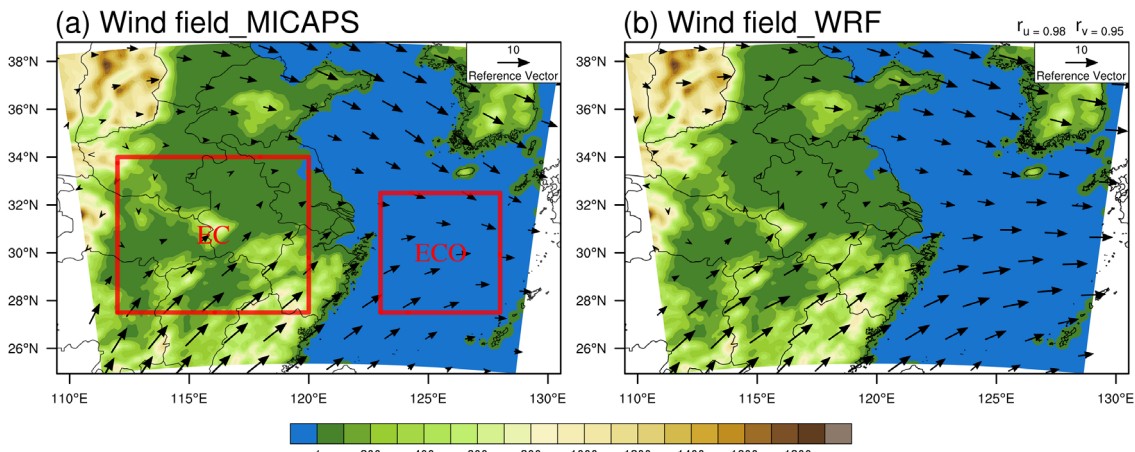

**Figure 2.** Topography (unit: m) of the model domain, MICAPS (a) and assimilated simulated (b) 850 hPa wind fields (unit: m·s⁻¹) during the simulation period and their correlation coefficients of u and v components ($r_u$、$r_v$) given in the upper right corner

We conducted three experiments based on the modified coupling system as well as the original model, namely SBM, SBM-NEW and MOR, which are described in Table 1. The physical settings are listed in Table 2. Furthermore, SBM-NEW and MOR experiments treat aerosol-chemical processes with CBMZ and 4-bin MOSAIC scheme (Sha et al., 2019), photolysis processes with Fast-J scheme (Wild et al., 2000), sea salt and dust emissions following MOSAIC/SORGAM and Zhao et al. (2010) respectively, as well as biogenic emissions with the MEGAN scheme (Guenther et al., 2006).

**Table 1.** Description of the experiments

| Name | Microphysics | Aerosol |
|---|---|---|
| SBM | Original SBM (bin scheme) | Default aerosol profile and simplified treatment |
| SBM-NEW | Modified SBM (bin scheme) | Modified MOSAIC online aerosol module |
| MOR | MORRISON (bulk scheme) | Original MOSAIC online aerosol module |



**Table 2.** Model settings of the physical parameterizations

| Physical process | Number | Name |
| --- | --- | --- |
| Longwave radiation | 4 | RRTMG (Mlawer et al., 1997) |
| Shortwave radiation | 3 | CAM (Collins et al., 2004) |
| Surface layer | 1 | MM5 Monin-Obukhov (Pahlow et al., 2001) |
| Land surface | 2 | Unified Noah (Chen et al., 2010) |
| Boundary layer | 1 | YSU (Shin et al., 2012) |
| Cumulus | 5 | Grell-3 (Grell and Freitas, 2014) |

## 2.3 Four-dimensional data assimilation

210    The accuracy of the meteorological field is crucial to reproduce realistic aerosol-cloud interaction scenarios, and thus a four-dimensional data assimilation approach is used to reduce the error of the simulated meteorological field. This approach utilizes relaxation terms based on the model error at observational stations to make the simulated meteorological fields closer to reality (Liu et al., 2005), thus exerting positive effects on the simulation of atmospheric physical and chemical processes (Rogers et al., 2013; Li et al., 2016; Ngan and Stein, 2017; Zhao et al., 2020; Hu et al., 2022). The data used for assimilation are obtained
215    from the NCEP operational global surface (https://rda.ucar.edu/datasets/ds461-0, last access: 19 March 2023) and upper-air (https://rda.ucar.edu/datasets/ds351-0, last access: 19 March 2023) observation subsets, which contain meteorological elements such as altitude, wind direction, wind speed, air pressure, temperature, dew point and relative humidity.

## 2.4 Observational data

We use multiple observations to assess the impact of the four-dimensional assimilation and the ability of the model to reproduce
220    meteorological fields, aerosols, and cloud parameters. The meteorological data are obtained from the Meteorological Information Combine Analysis and Process System (MICAPS) developed by the National Meteorological Center (NMC) of China (http://www.nmc.cn, last access: 19 March 2023), with 12 h temporal resolution and 11 vertical layers, containing meteorological elements such as wind field, geopotential height, temperature and temperature dew point difference. Near-surface $PM_{2.5}$ data are obtained from the National Urban Air Quality Real-time Release Platform of China National
225    Environmental Monitoring Centre with 1 h temporal resolution (https://air.cnemc.cn:18007, last access: 19 March 2023). The cloud parameters are obtained from the MODIS Level-2 Cloud (MOD06_L2) product (https://ladsweb.modaps.eosdis.nasa.gov/search/order/1/MOD06_L2--61, last access: 19 March 2023), from which we select cloud droplet effective radius (CER), cloud optical thickness (COT), CLWP and cloud phase data at 1 km resolution, as well as cloud top height (CTH), cloud top temperature (CTT) and cloud top pressure (CTP) at 5 km resolution. The CER, COT and
230    CLWP are retrieved from 2.1 μm wavelength, which is the default value in the product (1.6 μm and 3.7 μm wavelength



retrievals are also available).

Spatial correlation analysis, Pearson correlation analysis, and root mean square error (RMSE) are used to assess the spatial and temporal correlations of the simulated and observed values as well as the error of the simulated values relative to the observations. To calculate these parameters, it is necessary to unify the spatio-temporal coordinates of the simulated and observed data. We interpolate the observations into the same horizontal grid as the simulated data and interpolate the simulated data into the same vertical pressure layer as MICAPS when evaluating the meteorological field. For the MODIS data, we select the reliable cloud retrievals according to the approach of Saponaro et al. (2017): (1) selecting only liquid-phase cloud parameters and (2) filtering out transparent-cloudy pixels (COT < 5) to limit uncertainties (Zhang et al., 2012). The same filtering also applied to model outputs when doing evaluation against MODIS data. Since MODIS provides no information of $N_d$, we refer to the approach of Brenguier et al. (2000) and Quaas et al. (2006) utilizing MODIS COT and CER to calculate:

$$N_d = \gamma \cdot COT^{0.5} \cdot CER^{-2.5} \qquad (3)$$

where $\gamma$ is an empirical constant with the value of $1.37 \times 10^{-5}$ m$^{-5}$. Moreover, due to the discontinuity of MODIS data, we matched the simulated data with MODIS data in spatio-temporal coordinates for evaluation (i.e., the simulated value is valid only when the MODIS data is valid in that spatio-temporal coordinate, otherwise the simulated value is set as the missing and does not participate in the calculation).

## 3 Results and Discussion

### 3.1 Evaluation of assimilation effect

Four-dimensional data assimilation directly alters the meteorological field simulation and thereby affects the aerosol and cloud simulation. We first examine the effect of assimilation to clarify whether assimilation brings more confidence to the study. Figure 3 presents the vertical distribution of the simulated (SBM-NEW experiment) and observed meteorological elements before and after assimilation, as well as the RMSE of the simulated relative MICAPS observations at each layer. The four-dimensional assimilation exerts slight effect on the height field (Fig. 3a), which is almost absent at the lower layers and exhibits slight increase at the upper layers in the RMSEs with respect to the observations compared to the unassimilated. In contrast to the height field, the assimilation presents significant improvements to the simulated temperature (Fig. 3b), and the RMSEs are effectively reduced by assimilation at all layers. The effects of assimilation on the temperature dew point difference (Fig. 3c) and wind v component (Fig. 3e) exhibit reduced low and high layer RMSEs and enlarged middle layer RMSEs, while the effects on wind u component (Fig. 3d) exhibit reduced low and middle layer RMSEs and enlarged high layer RMSEs. As the complexity of atmospheric physical and chemical processes and data errors resulted from processes such as observation and station data gridding, the assimilation effects revealed by the evaluation are not uniformly positive, but overall exhibit positive effects. This positive effects are evident below 800 hPa, which especially helps to capture low clouds that dominate in winter.





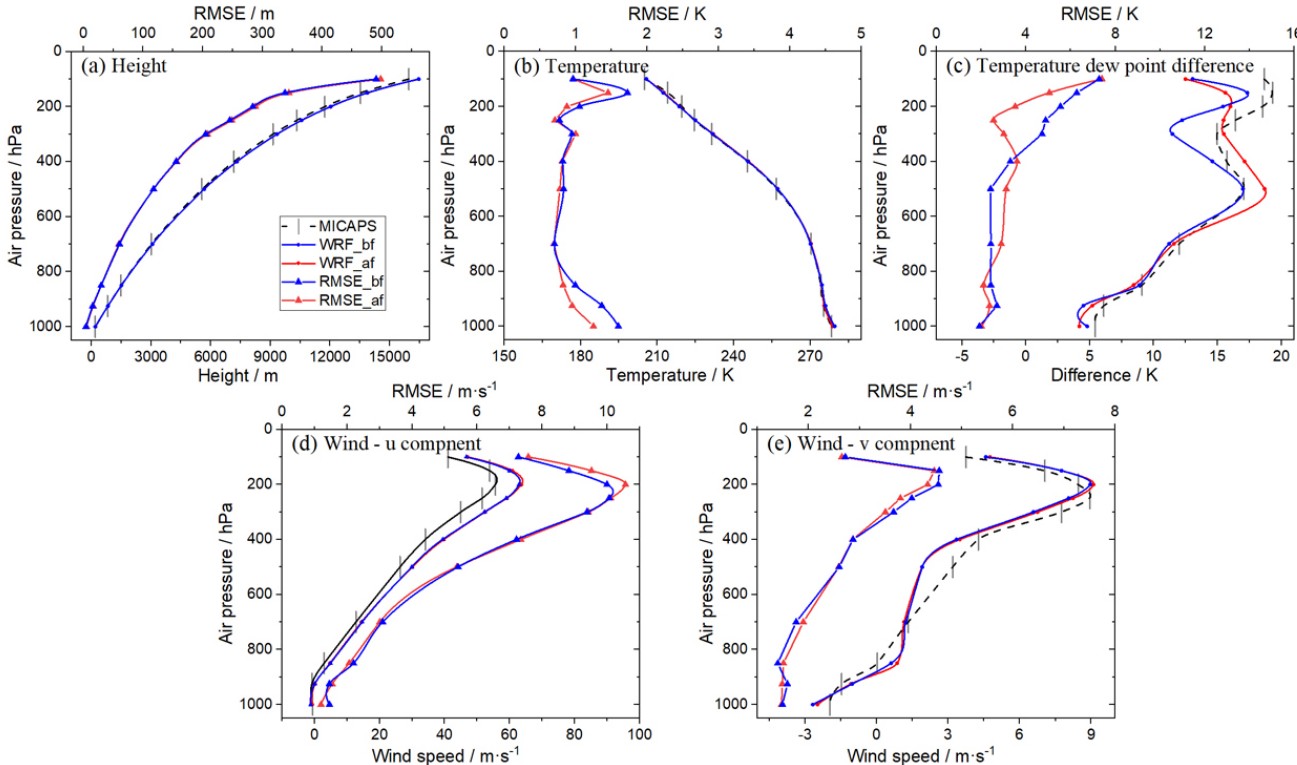

**Figure 3.** MICAPS and simulated height (a), temperature (b) and temperature dew point difference (c) as well as u (d) and v (e) components of wind and their RMSE vertical distribution before (blue line) and after (red line) assimilation

Assimilation modifies wind fields and precipitation, which in turn affects aerosol emissions (mainly natural aerosols such as dust and sea salt), transport and deposition. The simulated and observed near-surface $PM_{2.5}$ distributions are presented in Fig. 4. The simulated $PM_{2.5}$ before and after assimilation both reasonably reproduce the observed patterns, with the same spatial correlation coefficient of 0.95. However, the simulation without assimilation underestimates the PM2.5 concentration. Supported by assimilation, the model better reproduces the meteorological field as well as atmospheric physical and chemical

processes, thus effectively optimizes the aerosol simulation, with the RMSE between the simulated and observed near-surface $PM_{2.5}$ reduced from 30.7 µg m$^{-3}$ before assimilation to 24.1 µg m$^{-3}$. To evaluate the effect of assimilation on the simulation of $PM_{2.5}$ temporal variation, 16 stations with relatively continuous observation (Fig. 4b) are selected evenly from the model domain (Fig. 5). In general, the simulations before and after assimilation both reasonably reproduce the temporal variation of near-surface $PM_{2.5}$, and the correlation between simulated and observed $PM_{2.5}$ at all stations pass the test at 99% significance,

whereas the simulations before assimilation overall underestimate the $PM_{2.5}$ concentration. With assimilation, the simulated $PM_{2.5}$ concentrations are generally closer to the observations, and the correlation coefficients between the simulated and the observed have increased in 11 of the 16 stations, while the average correlation coefficient of the 16 stations has increased from 0.51 to 0.58.





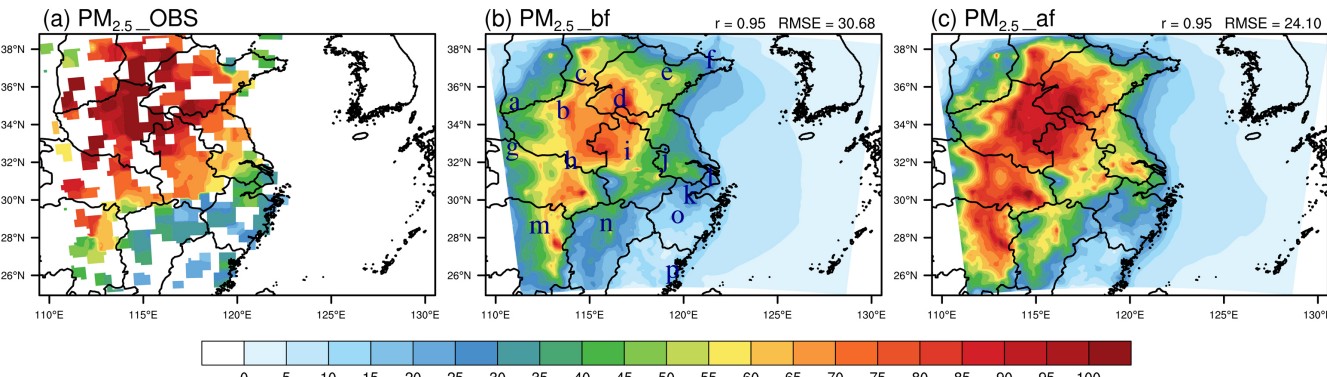

**Figure 4.** Distributions of average near-surface PM$_{2.5}$ (unit: µg·m$^{-3}$) during the simulation period from the observation (a) as well as before (b) and after (c) assimilation of the meteorological fields (r and RMSE at the up-right corner of Fig. b and c represent the spatial correlation coefficient and root mean square error of the observed and the simulated, respectively, where RMSE is in unit of µg·m$^{-3}$. The markers a-p in Fig. b represent the locations of the stations in Fig. 5)



**Figure 5.** Temporal variations of near-surface PM$_{2.5}$ observed (black line) and simulated before (blue line) and after (red line) assimilation of meteorological quantities, at each site. The r and p values represent the correlation and significance of the observation and simulation, respectively, and subscripts "bf" and "af" represent simulated before and after assimilation

As shown in the evaluation, the assimilation effectively enhances the model's ability to simulate meteorological fields
and aerosols, enabling better reproduction of real aerosol-cloud interactions.

### 3.2 Evaluation of the simulated cloud parameters

To investigate the differences in simulation performance among the SBM-MOSAIC coupling system, the original SBM (using the default aerosol profile) and the bulk microphysical scheme (MORRISON)-MOSAIC coupling system, we compare the cloud properties from three simulations (after assimilation) and MODIS during the simulation period.

Figure 6 presents MODIS and simulated CER (the CER of the SBM output is the quotient of the sum of the cubic and quadratic radius of the particles of each bin, and since the MORRISON scheme does not calculate CER, here do not include the MOR values), N$_d$ calculated from MODIS and simulated data based on empirical formula, and N$_d$ calculated directly by the model (the model outputs in the figure are spacing-weighted averages of the values for the vertical layers with clouds). For CER (Fig. 6a-c), the SBM experiment basically reproduces the MODIS distribution with a spatial correlation coefficient of
0.85, but the lack of reasonable aerosol distribution information results in a significant overestimation. By introducing online aerosol information, the SBM-NEW experiment effectively reduces this error (RMSE reduction by 43.6%) and further improves the spatial correlation between the simulated and MODIS CER (to 0.91). As MODIS does not provide N$_d$, MODIS N$_d$ is calculated by an empirical formula, which can provide information on the spatial distribution of N$_d$ to some extent but without sufficient accuracy. As the N$_d$ values of the SBM experiment shown in Fig. 6e and h, the calculated values exhibit an
overall overestimation of the simulated low values and underestimation of the simulated high values, which is attributed to this formula being mainly applicable to relatively homogeneous, optically thick and unobstructed stratiform clouds under high solar zenith angle conditions (Jia et al., 2021), which cannot be fully satisfied by the simulated and MODIS data used in this study. But in terms of the calculated MODIS N$_d$, the SBM-NEW experiment also exhibits the best reproducibility among the three experiments, with a correlation coefficient of 0.63 between the simulated and MODIS data, slightly lower than 0.64 for
the MOR but much higher than 0.23 for the SBM, and the RMSE with MODIS data (296.63 cm$^{-3}$) is much lower than that of the MOR (940.95 cm$^{-3}$) and the SBM (887.69 cm$^{-3}$).



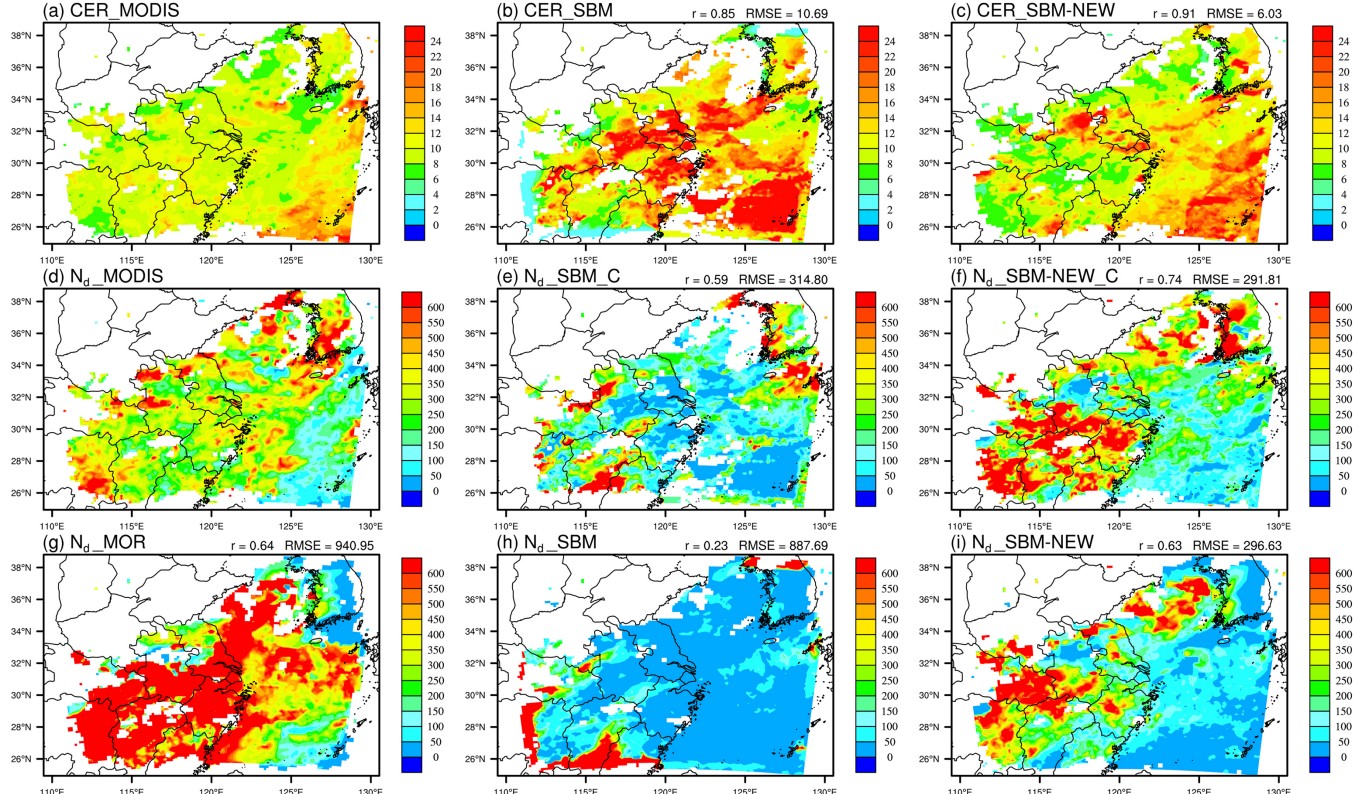

**Figure 6.** MODIS and simulated CER (in μm) and $N_d$ (in cm$^{-3}$) distributions (a, b and c are CER from MODIS, SBM and SBM-NEW. d, e and f are $N_d$ calculated using MODIS, SBM and SBM-NEW results based on empirical equation. g, h and i are $N_d$ from MOR, SBM and SBM-NEW simulations. r and RMSE on top right represent the spatial correlation coefficient and root mean square error of the simulated and MODIS data, where the RMSE of CER and $N_d$ are in μm and cm$^{-3}$, respectively)

For the physical schemes utilized in this study, the CLWP is the function of $N_d$ and cloud droplet size, while the COT (which refers to the liquid-phase cloud optical thickness alone in the study) is the function of CLWP and CER. Since the decreasing $N_d$ usually leads to the increasing CER with constant dynamical and thermal conditions in the model physics framework, the simulated CLWP and COT distributions in the SBM experiment are well in agreement with MODIS data despite the fact that the SBM fails to reasonably reproduce the $N_d$ distribution (Fig. 7), and even the simulated results are superior to the MOR coupled with online aerosol, which also reflects the advances of the bin scheme. By providing more reasonable aerosol information, the SBM-NEW has further improved the correlation (reach 0.84) and reduced the RMSE with MODIS compared to the SBM.

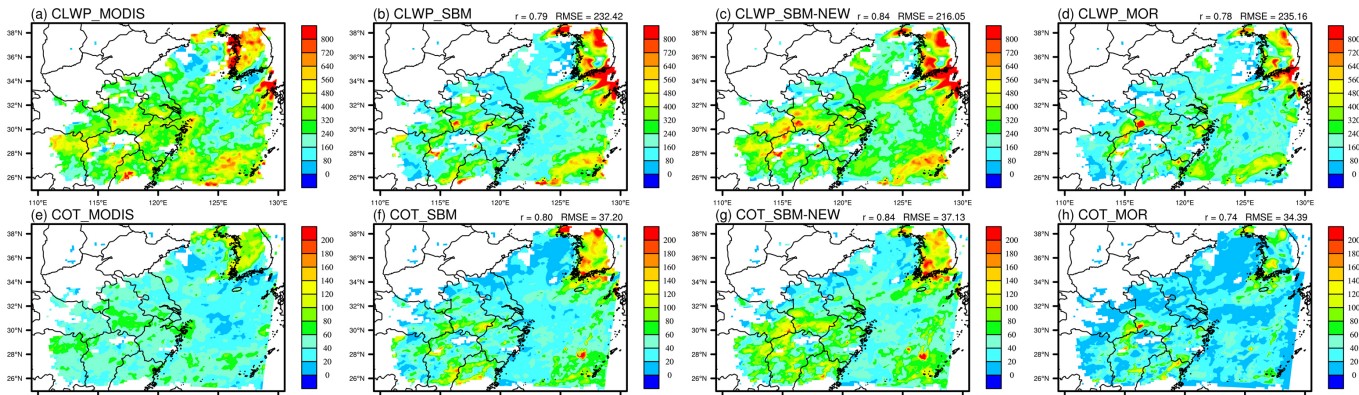

**Figure 7.** MODIS (a and e) as well as SBM (b and f), SBM-NEW (c and g) and MOR (d and h) simulated CWP (in g·m⁻³) and COD (dimensionless) distributions (The definitions of r and RMSE on top right are the same as in Fig. 6, where the RMSE of CLWP is in units of g·m⁻³)


In comparison with other cloud parameters, CTH, CTT and CTP are less sensitive to aerosol and mainly depend on dynamical and thermal conditions as well as macroscopic physical processes. So relying on the advancement of physical structure, the SBM simulates cloud top parameters with significantly higher accuracy than the MOR (Fig. 8), which is more obvious than for CLWP and COT. For the same reason that cloud top parameters are less sensitive to aerosol, the SBM-NEW

based on the same physical structure simulates cloud top parameters with little difference from the SBM, but the evaluation indicators still exhibit some improvement compared to the SBM.

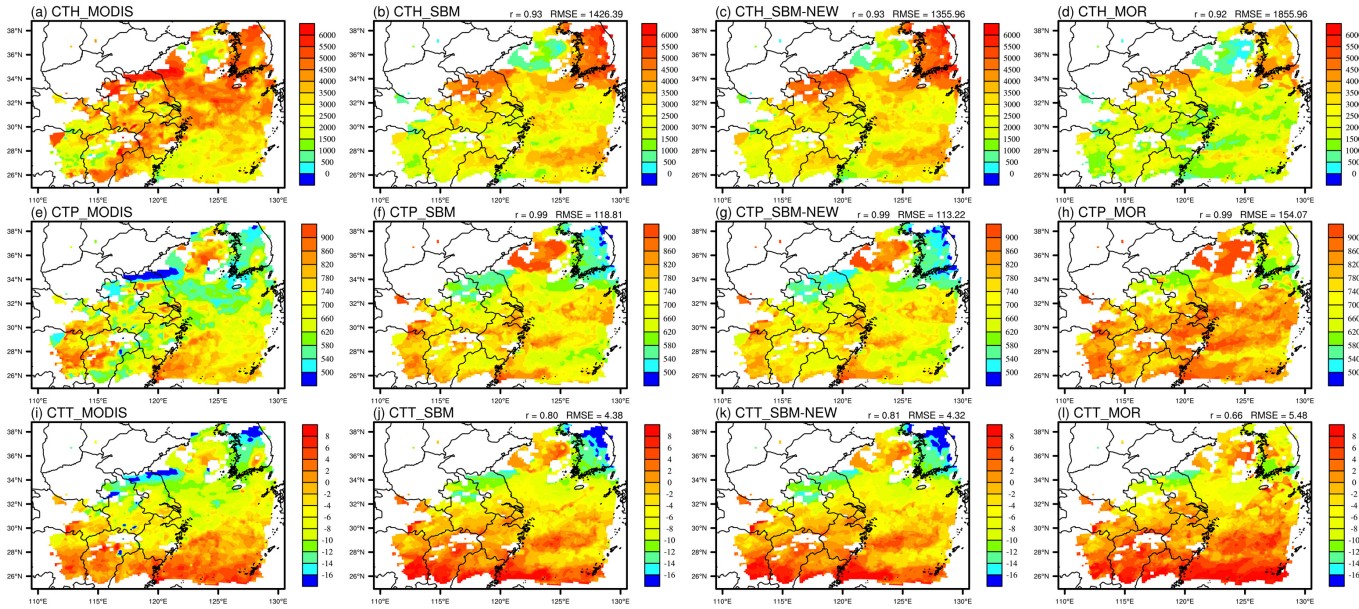





**Figure 8.** MODIS (a, e and i) as well as SBM (b, f and j), SBM-NEW (c, g and k) and MOR (d, h and l) simulated cloud top
height (in m), cloud top pressure (in hPa) and cloud top temperature (in °C) distributions (The definitions of r and RMSE on
top right are the same as in Fig. 6, where the RMSE of CTH, CTP and CTT are in m, hPa and °C, respectively)

In general, the SBM fails to reproduce $N_d$ and CER as its aerosol information is derived from the default profile (as shown
in Fig. S1), but the simulated macroscopic cloud parameters are in better agreement with MODIS than the MOR due to the
superiority of the bin scheme over the bulk scheme in terms of calculation accuracy. The SBM-NEW has better simulation
performance than the SBM and the MOR for both cloud microscopic and macroscopic parameters, which indicates that the
SBM-MOSAIC coupling system can better reproduce the physical and chemical processes in clouds and provide more realistic
and valuable information for the research of aerosol-cloud interactions. The following analysis will be based on the SBM-
NEW.

**3.3 Aerosol and cloud droplet distribution characteristics in EC and ECO**

The aerosol physical and chemical processes, aerosol-cloud interactions, and consequent aerosol and cloud droplet distribution
characteristics in EC and ECO exhibit distinct differences due to their differences in aerosol emissions and meteorological
fields. EC aerosols are mainly primary and secondary aerosols produced by anthropogenic emissions, with small initial particle
size. Limited atmospheric water content and excessive number of particles competing for water result in more but smaller
aerosol over EC. ECO aerosols in winter mainly comes from the transport of EC, making the majority are also small particles,
but since its locally emitted sea salt generally has larger sizes and atmospheric water is relatively abundant (relative humidities
are more elevated), the proportion of large particles is relatively high. As shown in Fig. 9, the number of ECO aerosols in the
fourth bin (1.25-5 μm in radius) is higher than that of EC despite a much lower total aerosol concentration. In addition, as the
ECO atmosphere is relatively clean, there are fewer particles competing for water, enabling a higher proportion of small
particles to be activated and better growth conditions for the droplets (The total $N_d$ of ECO is lower than that of EC due to
fewer aerosol particles, but the $N_d$ with more than 10 μm in radius is clearly higher than that of EC).



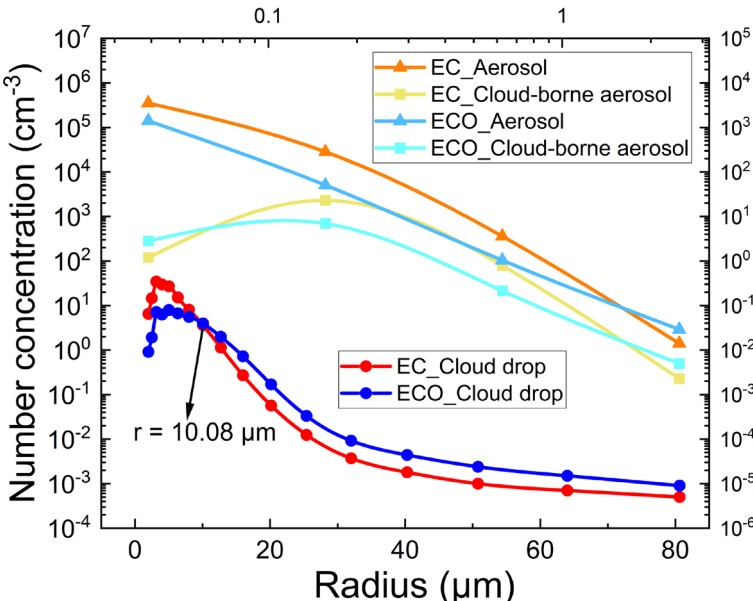

**Figure 9.** EC and ECO cloud droplets and particle size distribution of total and activated aerosols (lower and left axes correspond to cloud droplets, right and upper axes correspond to aerosols)


   Besides size distributions, EC and ECO also exhibit clear differences in the spatial distribution of aerosols and cloud droplets. EC aerosol mainly originates from surface emissions, so its concentration gradually decreases from surface to upper-air (Fig. 10a), while the ECO aerosol number concentration ($N_{aero}$) also exhibits an overall decreasing trend with increasing altitude, but since the important contribution of EC aerosol import, the $N_{aero}$ high value is located at the main transport altitude

of 1000-1500 m above sea level (Fig. 10e). Due to the longwave radiative cooling, a large amount of cloud droplets are distributed near the surface (i.e., fog) of EC, and the near-surface areas around 29°N and 31°N (Fig. 10b) exhibit high atmospheric supersaturation due to the effect of topographic uplift (Fig. 10c), by which the $N_d$ hotspots are generated. Because of the difference of underlying surfaces, there are almost no cloud droplets appearing at the ECO surface, and the area of high $N_d$ is located at 500-1500 m altitude where both aerosol concentrations from long-range transport are larger and supersaturation

due to turbulence is higher (Fig. 10e-g). Compared with ECO, EC features relatively strong vertical motion, which has important effects on its upper-air atmospheric supersaturation and aerosol activation (Fig. 10c-d), making the EC upper-air cloud droplet distributed more chaotic. In contrast, the ECO convection is weak in winter, relying mainly on large-scale cooling and humidifying to enable atmospheric supersaturation, and cloud droplet distribution is relatively uniform. It is worth mentioning that since the model integration step is 60 s and the output interval is 1 h, the actual activation process occurs with

a high probability at a certain time step between the hours rather than on hour. For better reflecting the relationship among aerosol, cloud droplet and supersaturation, the supersaturation in this study is the instantaneous value at the time of the latest activation process. In addition, to ensure the accuracy of the averaged supersaturation, if the same supersaturation value is





output for several consecutive times, it is recognized that supersaturation is not reached between these times and no activation occurs, and the supersaturation of these times (except for the first of the consecutive times) is set to 0 when averaging.


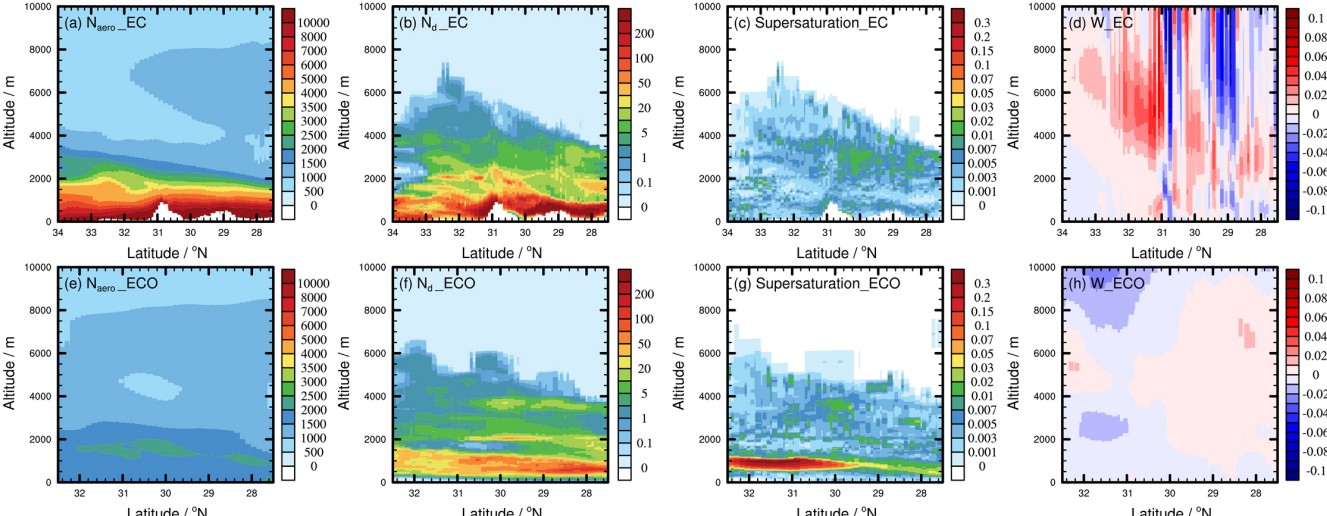

**Figure 10.** EC (a-d) and ECO (e-h) aerosol number concentration (in cm$^{-3}$), cloud droplet number concentration (in cm$^{-3}$), atmospheric supersaturation (in %) and vertical wind speed (in m·s$^{-1}$) distributions

### 3.4 Aerosol-cloud interaction signals in EC and ECO

The first step in aerosol-cloud interaction is aerosol activation, and we analyze the variation of $N_d$ with aerosol and its influencing factors based on the statistics of the model grids with CF greater than 0 at each time (Fig. 11). At low $N_{aero}$, aerosols boost cloud formation and development by acting as CCN, whereas at high $N_{aero}$, it is difficult for aerosols to grow into large particles because of limited atmospheric water content, and processes such as their hygroscopic growth make atmosphere harder to reach supersaturation, thus inhibiting activation process. So as shown in Fig. 11a-b, both EC and ECO $N_d$ exhibit the 395 variation characteristics of increasing first and then decreasing with increasing $N_{aero}$.



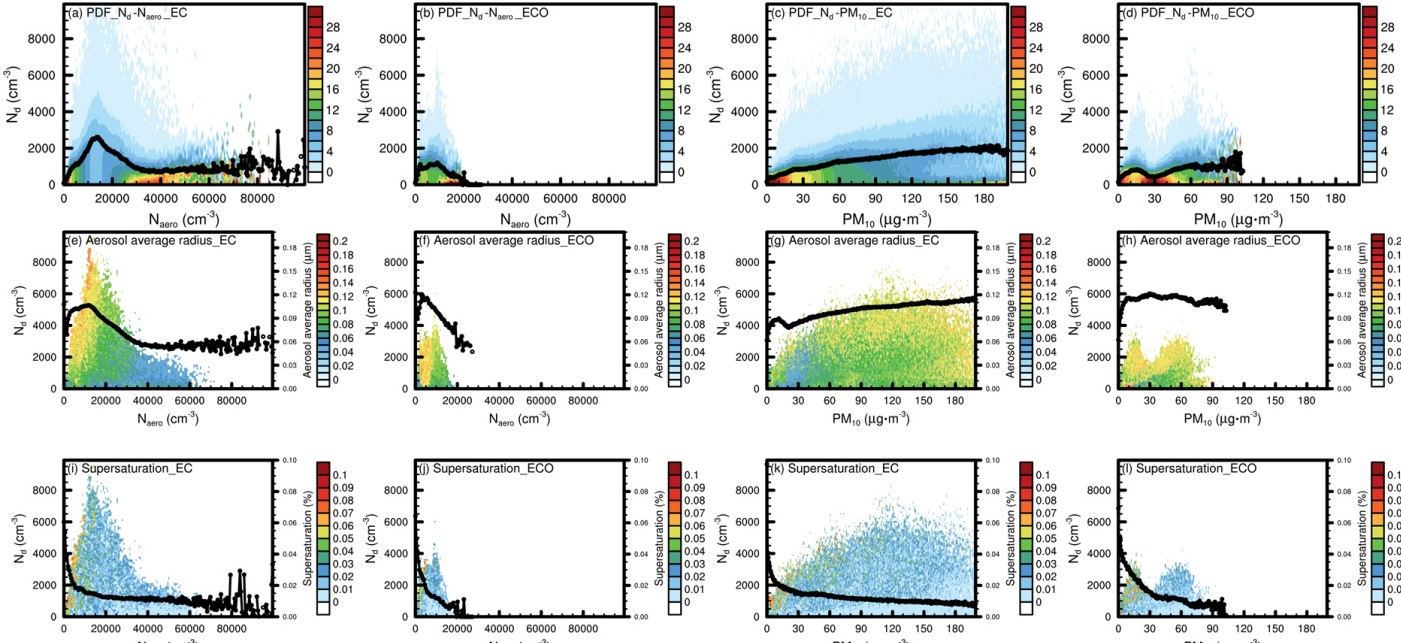

**Figure 11.** Probability density distribution functions (sum of probabilities corresponding to 1 for each $N_{aero}$ or $PM_{10}$ value) and means (lines in a-d) of the simulated $N_d$ relative to $N_{aero}$ (a-b) and $PM_{10}$ (c-d), as well as the values (corresponding to the left and bottom axis) of aerosol average radius (e-h) and supersaturation (i-l) along with their mean changes (lines in e-l, corresponding to the right and bottom axis) relative to $N_{aero}$ (e-f and i-j) and $PM_{10}$ (g-h and k-l) in EC (a, c, e, g, i, and k) and ECO (b, d, f, h, j, and l), respectively

During the simulation period, although the overall variations are quite similar, the $N_d$ variations in EC and ECO exhibit different characteristics in each $N_{aero}$ interval due to the discrepancies in aerosol and meteorological conditions. At the first stage, $N_{aero}$ is very low (0-2000 cm$^{-3}$), where both aerosol hygroscopic growth and activation are able to fully perform, as reflected by the rapid increases in the aerosol average radius (defined in this study as the radius corresponding to the average volume of aerosol particles, as shown in Fig. 11e-f) and $N_d$ (Fig. 11a-b) as well as a quick decrease in supersaturation (Fig. 11i-j). ECO aerosols at this stage come mainly from natural sources, which emit particles with larger size and lower activation threshold relative to anthropogenic aerosols, and the average growth rate of $N_d$ with $N_{aero}$ in ECO (average of 0.50 cm$^{-3}$·cm$^{-3}$) is much higher than that in EC (average of 0.24 cm$^{-3}$·cm$^{-3}$). The second stage ($N_{aero}$ value of 2000-10000 cm$^{-3}$), in EC with increasing aerosol, the competition of particles for water intensifies, and the growth of aerosol size slows down, but $N_d$ can still increase swiftly with $N_{aero}$ due to relatively strong updraft and surface radiative cooling enabling to guarantee the supersaturation required for aerosol activation. This stage in ECO mainly occurs at the altitude above 1000 m due to EC aerosol transport. As atmospheric water content decreases and major aerosol component starts to shift from natural aerosol with larger size to anthropogenic aerosol from EC with smaller size, aerosol average radius decreases rapidly. The absorption of water by



these small particles makes the atmosphere harder to reach supersaturation, and their smaller size results in higher supersaturation required for activation, slowing down the growth of ECO $N_d$ with $N_{aero}$ significantly. The third stage (EC $N_{aero}$ at 10000-30000 cm$^{-3}$ and ECO $N_{aero}$ at 10000-20000 cm$^{-3}$) EC and ECO exhibit similar characteristics, i.e., as $N_{aero}$ increases, either EC or ECO atmospheric moisture is insufficient to adequately supply the aerosol hygroscopic growth, aerosol average radius reduces promptly, excessive small particles absorb a large amount of water making it difficult to reach supersaturation in the atmosphere, and together with its higher activation supersaturation threshold, the aerosol activation is suppressed and $N_d$ decreases quickly. At the fourth stage (EC $N_{aero}$ above 30000 cm$^{-3}$ and ECO $N_{aero}$ above 20000 cm$^{-3}$), the EC $N_d$ tends to a constant (about 800 cm$^{-3}$) with increasing $N_{aero}$. As $N_{aero}$ increases further above 60000 cm$^{-3}$, $N_d$ exhibits sharp fluctuations, which generally appear near the surface and are mainly attributed to diurnal variations, i.e., strong surface radiative cooling effect at night or early morning leads to significant increases in atmospheric supersaturation, which in turn promotes aerosol activation and $N_d$ increase, while the absence of surface cooling in daytime results in rapid reduction of the increase rate of $N_d$ with $N_{aero}$. Compared to EC, ECO in winter has neither surface cooling nor updraft of adequate strength, and aerosol activation and cloud droplet generation can hardly happen after $N_{aero}$ above 20000 cm$^{-3}$. Dissimilar to the variation with $N_{aero}$, both EC and ECO $N_d$ increase with PM$_{10}$ (Fig. 11c-d), attributed to high PM$_{10}$ values dominated by larger particles that are more prone to activate (Fig. 11g-h).

Aerosol activation leads to altered cloud droplet size distribution and consequent changes in cloud microphysical and dynamical processes, which is also known as rapid adjustment (Heyn et al., 2017; Mulmenstadt and Feingold, 2018). We discuss the variations of cloud liquid water content (CLWC) and CER in EC and ECO with increasing $N_d$ for both precipitation clouds (raindrop number concentration above 500 m$^{-3}$) and non-precipitation clouds (raindrop number concentration of 0 m$^{-3}$). Aerosols entering clouds usually lead to more but smaller cloud droplets and suppress the growth of cloud droplets into raindrops, which in turn results in reduced precipitation efficiency, increased cloud water content and CF, as well as longer cloud lifetime, which is the first rapid adjustment mechanism, also known as the second aerosol indirect effect (Albrecht, 1989). This effect dominates in the precipitation cloud, as shown in Fig. 12a-b and e-f. Both CLWC and CER show stable increasing and decreasing trends with increasing $N_{aero}$ for the precipitation cloud in EC and ECO, respectively, but there exist some differences between the two regions. At low $N_d$ in EC (0-130 cm$^{-3}$), strong updrafts and radiative cooling of surface provide superior growth conditions for cloud droplets, enabling CLWC to increase rapidly with $N_d$ (mean increase rate of $1.7 \times 10^{-3}$ g·m$^{-3}$·cm$^{-3}$). However, as $N_d$ increases, not only cloud droplets compete for water, but also processes such as hygroscopic growth of aerosols consume water, leading to slower increase of CLWC with $N_d$ and the increase rate tends to a constant ($3.4 \times 10^{-4}$ g·m$^{-3}$·cm$^{-3}$), and CER reduces gradually with $N_d$ and tends to 5 μm. Due to the different ways of supersaturation from EC, ECO does not show similar CLWC outburst increase at very low $N_d$, nevertheless, it exhibits fast CLWC increase rate (average of $8.3 \times 10^{-4}$ g·m$^3$·cm$^{-3}$) when $N_d$ is lower than 1300 cm$^{-3}$ because it contains relatively abundant water for aerosol hygroscopic growth and the locally emitted natural aerosol has larger initial size. But as $N_d$ increases, i.e., a large amount of aerosol transported from EC, ECO can neither supply enough water for aerosol hygroscopic growth nor sufficient supersaturation, and its CLWC increases slowly with $N_d$ and tends to a much lower average increase rate ($8.0 \times 10^{-5}$ g·m$^{-3}$·cm$^{-}$



[3] in the $N_d$ range of 3600-5000 $cm^{-3}$) than that in EC. Also due to relatively abundant water and higher percentage of natural aerosols, CER in ECO decreases slower than EC with $N_d$, but also gradually converges to 5 μm with the numerous input of anthropogenic aerosols and the lack of water supply. For non-precipitation clouds, the variations of CLWC and CER with $N_d$ in EC and ECO exhibit more significant differences. The variation of CLWC with $N_d$ in EC mainly relies on atmospheric supersaturation due to updrafts and surface radiative cooling, so compared with precipitation clouds, non-precipitation clouds

can also maintain a stable increase in CLWC with $N_d$, although their water content is lower and the increase is relatively slow. While the variation of CLWC with $N_d$ in ECO is mainly dominated by available water content, when there are few cloud droplets and the aerosol is mainly naturally emitted large particles, CLWC can still increase rapidly with $N_d$, but with the numerous input of small particles from EC, ECO CER decreases rapidly and the second rapid adjustment mechanism (i. e., increased $N_d$, reduced CER and harder deposition leading to additional cloud top cooling and further driving entrainment and

evaporation, as well as increased surface area to volume ratio of cloud droplets due to reduced CER, further enhancing evaporation and cloud top negative buoyancy) begins to prevail, making CLWC decrease with increasing $N_d$.

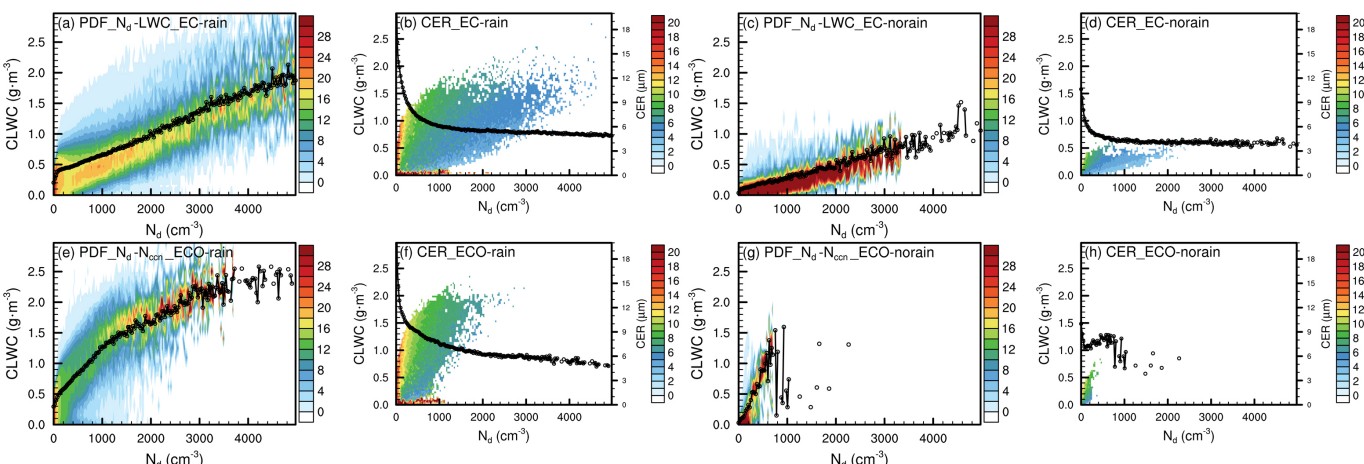

**Figure 12.** Probability density distribution functions (sum of probabilities corresponding to 1 for each $N_d$ value) and means

(lines in the figures) of CLWC relative to $N_d$ (a, c, e, and g), as well as the CER values (b, d, f, and h, corresponding to the left and bottom axis) and their mean changes relative to $N_d$ (lines in the figures, corresponding to the right and bottom axis) of precipitation clouds (a-b and e-f) and non-precipitation clouds (c-d and g-h) in EC (a-d) and ECO (e-h)

In addition to the above analysis, we note the anomaly of few CLWC at high CER and $N_d$ in Fig. 12b and f ($N_d$ values lie

in the range of 0-2200 $cm^{-3}$ and CLWC values in the range of 0-0.1 $g \cdot m^{-3}$). It is attributed to the asynchronicity caused by the fact that CER is calculated and output directly in the SBM scheme only, while $N_d$ and CLWC are involved in the subsequent physical and chemical process treatment and may be changed. This asynchrony appears quite infrequently (as presented by the probability density distribution shown in Fig. 12a and e) and poses minor impact on long-term mean state and general tendency



analysis, but may cause some disturbance to short-term analysis of individual cases. Moreover, both the SBM-MOSAIC

coupling system used in this study and the bulk-MOSAIC coupling system in the original WRF-Chem use the threshold
approach to calculate CF, i.e., CF equals to 1 when the sum of cloud water and cloud ice mixing ratios is greater than $10^{-6}$
$kg \cdot kg^{-1}$, otherwise equals to 0. Although we chose a more refined CF calculation approach (set icloud to 1) in the model to
involved in the physical process treatment, since the CF calculation of the aerosol-chemistry module is called later, its result
will overwrite the value calculated in the previous physical treatment, which makes us unable to make further analysis of the

aerosol-CF responses. We will further optimize the model in subsequent studies to solve these issues and provide stronger
support for the study of aerosol-cloud interactions.

## 4 Conclusion

In this study, aerosol-cloud physical and chemical processes are treated more precisely by coupling the spectral-bin cloud
microphysics (SBM) and online aerosol module (MOSAIC) in WRF-Chem model, and aerosol-cloud interactions in eastern

China (EC) and its adjacent ocean region (ECO) in winter are explored based on the coupling system and the four-dimensional
assimilation approach. The coupling is structured by (1) adjusting the CCN bins of SBM to interface with the MOSAIC aerosol,
(2) treating aerosol activation, resuspension, and in-cloud removal processes at SBM instead of MOSAIC, and (3) calculating
the parameters required for other wet removal treatments by SBM and passing them to MOSAIC.

We evaluated the impact of four-dimensional data assimilation and the simulation ability of the coupling system using

multiple observations. The evaluation indicates that the assimilation exhibits an overall positive impact on the simulation, but
the discrepancy between the simulated and observations is enlarged for some elements due to the complexity of the atmospheric
physical and chemical processes and the errors arising from the observation and station data gridding. Specifically, assimilation
optimizes temperature simulations at all vertical layers, with both positive and negative effects on temperature dew point
differences and wind fields at each layer, and little effect on height field, with consequent reductions of 21.4% in RMSE and

improvements of 13.7% in temporal correlation coefficient of the simulated relative observed near-surface $PM_{2.5}$. By coupling
SBM with the online aerosol module, MOSAIC provides SBM with more reasonable aerosol information, and SBM feedback
MOSAIC with more physical treating of aerosol in-cloud processes and more accurate microphysical parameters, their mutual
promotion makes the simulation ability of SBM-MOSAIC coupling system significantly improved compared with the original
SBM and bulk-MOSAIC coupling system, and each satellite-retrieved cloud parameters are reasonably reproduced in its

simulation results. This evaluation also provides a basis for confidence in the aerosol-cloud interactions exhibited by SBM-
MOSAIC coupling system.

Due to the differences in aerosol composition and meteorological fields, the aerosol and cloud droplet distribution
characteristics in EC and ECO exhibit clear discrepancies in winter. The majority of EC aerosols come from anthropogenic
emissions, with excessive quantities and poor atmospheric water content resulting in small aerosol particle sizes and high

activation supersaturation thresholds. The main pathways of atmospheric supersaturation over EC are radiative cooling at



surface, topographic uplift and updraft, and cloud droplets are mainly produced in the radiative cooling and uplift areas in the near-surface region with high aerosol content and in the area of strong updraft with disorderly distribution at high altitude. ECO aerosols mainly come from local natural emissions and EC transport, and relatively high proportion of natural aerosols and abundant water make it have larger particles than EC when the total amount is much less than EC. In winter, the updraft
over ECO is very weak, and the main pathway of atmospheric supersaturation is large-scale cooling and humidifying. Cloud droplets in ECO mainly appear at 500-1500 m altitude where the aerosol and supersaturation conditions are most favorable, and the overall distribution is more uniform.

Aerosol activation is dominated by aerosol number and size as well as atmospheric water content. When aerosol particles are quite few (0-2000 cm$^{-3}$), both aerosol hygroscopic growth and activation can be fully performed, and cloud droplet number
concentration ($N_d$) increases rapidly with aerosol number concentration ($N_{aero}$) after the atmosphere reaches supersaturation, with the average growth rate of $N_d$ with $N_{aero}$ being much higher in ECO (0.50 cm$^{-3}$·cm$^{-3}$) than in EC (0.24 cm$^{-3}$·cm$^{-3}$) due to higher proportion of naturally emitted large particles. As $N_{aero}$ increases (2000-10000 cm$^{-3}$), particle competition for water intensifies and the stronger route to supersaturation in EC allows it to still satisfy aerosol hygroscopic growth and activation, while ECO decreases in average particle size and atmospheric water content due to the input of numerous anthropogenic
aerosols from EC, and $N_d$ increases slow down substantially with $N_{aero}$. When $N_{aero}$ increases further, neither EC nor ECO atmospheric water is sufficient to adequately supply aerosol hygroscopic growth. Excessive small particle uptake of water and a higher activation supersaturation threshold suppress the activation process, causing $N_d$ to decrease rapidly with $N_{aero}$. The difference is that the $N_d$ in EC fluctuates around 800 cm$^{-3}$ due to strong atmospheric supersaturation caused by the radiative cooling effect of the surface at night or early morning, while activation in ECO is gradually suppressed and almost no cloud
droplets are produced. Unlike the variation with $N_{aero}$, $N_d$ with PM$_{10}$ in both EC and ECO increase due to high PM$_{10}$ values dominated by larger particles that are more readily activated.

We discuss the rapid adjustments in EC and ECO for two cases: precipitation clouds (raindrop number concentration higher than 500 m$^{-3}$) and non-precipitation clouds (raindrop number concentration of 0 m$^{-3}$). Both CLWC and CER show stable increase and decrease trends with $N_{aero}$ in precipitation clouds regardless of EC or ECO, respectively. The difference is that
the more bursty atmospheric supersaturation and subsequent lack of water content in EC lead to an explosive increase of CLWC with $N_d$ at an average rate of $1.7 \times 10^{-3}$ g·m$^{-3}$·cm$^{-3}$ when there are few cloud droplets ($N_d$ values of 0-130 cm$^{-3}$), but the increase quickly slows down to $3.4 \times 10^{-4}$ g·m$^{-3}$·cm$^{-3}$ when $N_d$ is above 130 cm$^{-3}$. ECO, owing to its large scale mild atmospheric supersaturation path, exhibits fast increase (average rate of $8.3 \times 10^{-4}$ g·m$^{-3}$·cm$^{-3}$) in CLWC at $N_d$ range 0-1300 cm$^{-3}$ with high proportion of large aerosol particles, but as $N_d$ increases, i.e., numerous input of aerosols from EC, ECO can neither supply
enough water for aerosol hygroscopic growth nor sufficient supersaturation, and its CLWC increase gradually slows down with $N_d$ and tends to a very low rate (average of $8.0 \times 10^{-5}$ g·m$^{-3}$·cm$^{-3}$ at $N_d$ range of 3600-5000 cm$^{-3}$). For non-precipitation clouds, EC and ECO exhibit more clear discrepancies. Because the aerosol-cloud processes depend mainly on the high supersaturation produced by atmospheric vertical motion and surface effects, the CLWC in EC non-precipitation clouds increases slowly with $N_d$ relative to precipitation clouds, but can maintain a stable growth trend. Whereas in ECO, which relies



mainly on large scale water and temperature variations to reach supersaturation, for non-precipitation clouds with less water content, the size of cloud droplets decreases and deposition becomes harder with increasing $N_d$, and further enhances entrainment and evaporation as well as negative buoyancy at cloud top, leading to reduction in CLWC.

*Code availability.* The code used in this study is modified from WRF-Chem, which is available at
https://www2.mmm.ucar.edu/wrf/users/download/get_sources.html (last access: 19 March 2023). The modified code will be available upon request.

*Data availability.* The model outputs are available upon request, the other data can be accessed from the websites listed in Sect. 2.

*Author contributions.* JZ and XM designed and conducted the model experiments, analysed the result and wrote the paper. XM developed the project idea and supervised the project. XM, JQ and HJ proposed scientific suggestions and revised the paper.

*Competing interests.* The authors declare that they have no conflict of interest.

*Acknowledgements.* This study is supported by the National Natural Science Foundation of China (Grants 42061134009 &41975002) and the Postgraduate Research and Practice Innovation Program of Jiangsu Province (Grant KYCX22_1151). The numerical calculations in this paper was conducted in the High-Performance Computing Center of Nanjing University of
Information Science & Technology. We are grateful to the National Aeronautics and Space Administration, the National Center for Environmental Prediction, MEIC Support Team, the Chinese National Meteorological Center and China National Environmental Monitoring Centre for providing the MODIS data, FNL and observation subsets, MEIC emission inventory, MICAPS data and $PM_{2.5}$ data respectively.

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
