# Peer review of "Exploring aerosol-cloud interactions over eastern China and its adjacent ocean using the WRF-SBM-MOSAIC model"

_EGUsphere, 2023_

## Author Comment (AC1)

**Response to the Comments of Referees**

**Journal:** Atmospheric Chemistry and Physics
**Manuscript Number:** egusphere-2023-331

**Title:** Exploring aerosol-cloud interactions over eastern China and its adjacent ocean using the WRF-SBM-MOSAIC model

**Author(s):** Jianqi Zhao, Xiaoyan Ma, Johannes Quaas, and Hailing Jia

We thank the reviewers and editor for providing helpful comments to improve the manuscript. We have revised the manuscript according to the comments and suggestions of the referees.

The referee's comments are reproduced (black) along with our replies (blue). All the authors have read the revised manuscript and agreed with the submission in its revised form.

**Anonymous Referee #1**

Review of "Exploring aerosol-cloud interactions over eastern China and its adjacent ocean using the WRF-SBM-MOSAIC model" by Zhao et al.

The study coupled WRF-Chem with SBM and used it to study aerosol-cloud interactions for a stratiform cloud case at 15-km resolution. There are a few serious problems as detailed below. Here is a high-level summary: (1) there is no clear motivation, particularly in terms of coupling MOSIAC with SBM since this was done and applied to many studies already, (2) the coupling that the authors did is not correct, (3) applying supersaturation-based activation in SBM to 15 km resolution is not appropriate. These are serious methodology problems, so I recommend a rejection of the study.

We thank the reviewer for taking the time to assess the manuscript and providing helpful comments and suggestions to improve the manuscript. Our point-by-point responses are as below: (1) we acknowledged that some previous studies have coupled SBM and MOSAIC, but such version is not available to public for use, and currently we don't see a version with the coupling of SBM and MOSAIC from the latest public versions of WRF-Chem. To better understand the physical mechanisms of aerosol-cloud interaction from modelling studies, it is necessary for us to couple SBM-MOSAIC in WRF-Chem since both the detailed cloud microphysics including cloud droplet size and aerosol microphysics (for example aerosol size distribution and chemical component) are key factors. We have included more statement to clarify our motivation in the revised manuscript. (2) The coupling system used in our submitted manuscript was based on particle size and ignored the differences in hygroscopicity of different aerosols. According to the reviewer's suggestion, we have imported the MOSAIC default hygroscopicity for each aerosol into SBM and treated the activation of each species of aerosol separately in order to make aerosol activation parameterization capable of resolving aerosol species. (3) This study focuses on liquid-phase clouds with wide range and long duration, and mainly investigates the general characteristics of aerosol-cloud variations rather than a specific process, so a larger temporal and spatial range needs to be adopted, and only a relatively coarse spatial and temporal resolution can be used under limited computing power. Although at 15km resolution mostly using bulk cloud microphysical scheme, our evaluation in Sect. 3.2 shows that the SBM has better simulation performance than the bulk scheme at 15 km resolution for both macrophysical and microphysical cloud properties. In addition, two sets of tests (see response to detailed comments below) are performed to examine the SBM simulation ability differences between the simulations of coarse temporal and spatial resolutions compared to fine resolutions. The test results show that the use of coarse temporal resolution only differs somewhat from the fine resolution simulations in the variation of $N_d$ due to the

difference in resolving meteorological fields, but in general both show consistent variation and do not cause neglect of aerosol activation and clear underestimation or overestimation of $N_d$ by using coarse resolution. In addition, the use of coarse spatio-temporal resolution shows consistent characteristics in both the spatial distribution of aerosol and cloud parameters and the variations of cloud parameters with aerosol. Only the lack of accuracy in resolving the meteorological field leads to relatively few strong activation samples and relatively low average $N_d$, which can be compensated by 1) optimization of the meteorological field simulation by four-dimensional assimilation, 2) resolving of sub-grid convection by cumulus parameterization,3) inclusion of more abundant atmospheric processes at a larger spatial and temporal scale, 4) selection of winter period dominated by cloud types with large temporal and spatial ranges such as stratocumulus, and 5) filtering out of ice-phase processes when analyzing. Due to a) the better simulation performance of SBM compared to the bulk scheme for cloud macroscopic and microscopic properties at 15 km resolution demonstrated in Section 3.2, b) the consistency of the simulated aerosol and cloud variations at coarse resolution with fine resolution revealed by the tests, c) the aforementioned measures to compensate for the lack of meteorological field simulation ability at coarse resolution, and d) the convenience of SBM for resolving aerosol particle size, aerosol composition and droplet size on aerosol-cloud interaction, it is appropriate to use 15 km resolution in this study. Please see the following detailed point-by-point responses.

**Detailed comments:**

Abstract:

1.  The advantage of WRF-Chem is to study aerosol effects from aerosol properties. It would allow us physically to study how $N_d$ is affected by different aerosol properties. However, the study did not take this advantage and they only examine the relationship of cloud properties with $N_d$ but did not even connect with aerosol properties. Aerosol effects starts from how $N_d$ are changed through aerosol properties.

    In Sec. 3.3, we examined the response of cloud droplet size spectra to aerosol size (Fig.9) in the submitted manuscript. In addition, we have included the analysis of response of cloud droplet to aerosol composition (Fig.10) in the revised manuscript. The detailed analysis and discussions are presented in Sec.3.3 and 3.4.

2.  The sentence "more bursty atmospheric supersaturation and lack of subsequent water cause cloud liquid water content (CLWC) in EC to increase explosively with $N_d$", the sentence is confusing. How would lack of water contribute to increase $N_d$?

    Thanks for the reminder. What we meant is that when water content is limited, CLWC exhibits rapid increase with $N_d$ if there are few cloud droplets, and then slow down. We have modified this sentence for clarification in the revised manuscript.

Introduction,

1.  The introduction is all about aerosol impacts on stratocumulus and warm clouds. Is this the cloud you study in this work? If so, you need to clarify this at the beginning of the introduction so that people would know your target since the mechanisms of aerosol effects are very different with different types of clouds.

    Yes, our study focus on liquid-phase cloud. We have clarified this in the revised manuscript.

2.  Line 81-82, "WRF-Chem currently only provides the coupling of bulk microphysical schemes with an online aerosol module (Gao et al., 2016)": This is totally wrong, what Gao et al. 2016 documented is the WRF-Chem coupled with spectral-bin microphysics (SBM). I noticed this is later stated. But here correction is needed.

    Corrected.

3.  Line 85-90, these statements were good for the motivation of Gao et al., (2016) because bin microphysics scheme was never coupled with chemistry/aerosol module before. However, it does not apply anymore now since Gao et al., (2016) already built the capability and it has been used in many studies such as Fan et al., 2020, https://doi.org/10.5194/acp-20-14163-2020, Zhang et al., 2021, https://doi.org/10.5194/acp-21-2363-2021.; Lin et al., 2021https://doi.org/10.1175/JAS-D-20-0106.1, , Lin et al. 2022https://doi.org/10.5194/acp-22-6749-2022). Thus I have a difficulty to understand the motivation of this work, which repeats Gao et al. (2016) without a justification.

    We acknowledged that some previous studies have coupled SBM and MOSAIC, and used it to examine aerosol-cloud interaction, specifically, for convective clouds. Some good examples are listed above by the reviewer. In our study, we focus on liquid-phase cloud, and attempt to explore how liquid-phase clouds response to aerosol physical properties as well as meteorological conditions. As we understand, although bin microphysics scheme was coupled with chemistry/aerosol module and also used in many studies, but such model version is not available to public for use, and currently we don't see a version from the latest public versions of WRF-Chem. To better understand the physical mechanisms of aerosol-cloud interaction from modelling studies, it is necessary for us to couple SBM-MOSAIC in WRF-Chem since both the detailed cloud microphysics including cloud droplet size and aerosol microphysics (for example aerosol size distribution and chemical component) are key factors.

Section 2,

1.  I am very confused by the writing of this section. It starts from describing "This study is based on WRF-Chem v3.9, the full version of the SBM scheme coupled with the aerosol module (Khain et al.,2009)", and provided detailed description of SBM, then saying SBM is not coupled with chemistry/aerosols instead of using prescribed CCN. First, in the introduction, Gao et al., 2016 which coupled with SBM with WRF-Chem is described. This writing is not only misleading but I'd ask what's the logic and point you want to deliver? The authors are totally ignoring Gao et al., 2016 here and repeating what was done in Gao et al. 2016 but did not provide any justification why you are doing this.

    Thanks for the comment. What we wanted to express is that current public versions of WRF-Chem model, even the recent WRF-Chem v4.4, SBM still uses the prescribed CCN, so we need to couple it with online aerosol in order to improve its ability to simulate aerosol-cloud processes. The original text is somewhat misrepresented. We have revised "This study is based on WRF-Chem v3.9, the full version of the SBM scheme coupled with the aerosol module (Khain et al.,2009)" to "This study is based on WRF-Chem v3.9, we use the full version of SBM (Khain et al., 2009) to couple with MOSAIC" and revised "In the current SBM scheme of WRF, the CCN…" to "In the current SBM scheme of the public versions of WRF-Chem, the CCN…". In addition, in the revised manuscript, we use an improved version of the aerosol activation parameterization compared to the previous manuscript, as detailed in the response to the next comment.

2.  Page 5-6, there is a significant problem I see in the coupling SBM with MOSAIC. MOSAIC

predicted various aerosol composition and hygroscopicity over different size ranges and it is physically wrong to map the aerosols from MOSIAC into aerosol bins in SBM based on size only, since activation of aerosols in SBM is only based on simple composition (default sea salt) to get the critical supersaturations for each bin. This is the key but most tricky part for this coupling. That is why Gao et al. (2016) implemented an interface code to set up the 33 critical supersaturations bins so that aerosols in the interstitial size bins in MOSAIC are mapped (i.e., distributed) to the 33 preset aerosols size bins through the interface module, and the mapping is based on particle critical supersaturations calculated from MOSAIC-predicted aerosol properties (size and hygroscopicity) rather than dry size. This was clearly described in Gao et al., 2016 and provided clear explanation why it is done in this way, i.e., "Since MOSAIC-predicted aerosol compositions vary with bins and over each bin a lognormal size distribution is assumed, mapping of aerosols from MOSAIC into SBM bins based on $S_{crit}$ is the easiest and most precise way to connect aerosols between the two models. That is, all aerosol types are represented on a single set of 33 CCN bins, but each aerosol type may distribute over different CCN sizes." Besides this, I see the authors follow Gao et al. 2016 on all other coupling treatments.

Thanks for the suggestion. The hygroscopicity is very important and was neglected in our submitted manuscript. In the SBM scheme, the aerosol activation parameterization is based on Kohler theory, which relies on the critical activation radius of aerosol calculated based on aerosol hygroscopicity and other parameters to treat aerosol activation. We have modified the coupling system based on the differences of aerosol hygroscopicity (the specific values are referred to the default values of the MOSIAC aerosol module, making the coupling system compatible) so that SBM can handle each aerosol composition and particle size, as described in lines 164-171 of the manuscript.

3. They use 15 km resolution for this study. The advantage of using SBM is diminished at such coarse resolution. On the contrary, the original WRF-Chem with the bulk schemes is more appropriate since supersaturations cannot be resolved much so activation is parameterized with Abdul-Razzak and Ghan parameterization. However, the activation of aerosols in SBM is based on supersaturations (not parameterized as Abdul-Razzak and Ghan scheme) but supersaturation would be poorly simulated at 15 km resolution (only very limited supersaturation can be resolved at such a coarse resolution). Therefore, this makes the effort of coupling SBM with WRF-Chem and the use of such physics-complicated model meaningless

Thanks for the comment. In previous studies, SBM was mostly used for individual convective system simulations, and the spatial and temporal resolution were mostly 1-3 km and 6-18 s. Our study focuses on liquid-phase clouds with wide range and long duration, and mainly investigates the general characteristics of aerosol-cloud variations rather than a specific process, so a larger temporal and spatial range needs to be adopted, and only a relatively coarse spatial and temporal resolution can be used under limited computing power. Although at 15km resolution mostly using bulk cloud microphysical scheme, our evaluation in Sec. 3.2 shows that the SBM has clearly superior simulation performance than the bulk scheme at 15 km resolution for both macrophysical and microphysical cloud parameters.

In addition, to understand the simulation ability of SBM at coarse resolution compared to fine resolution, we conducted two sensitivity tests trying to answer below questions: (1) whether coarse temporal resolution ignores many activation processes and causes overall aerosol-cloud-water

vapor simulation errors, and (2) whether reasonable aerosol-cloud variation information can be simulated at coarse spatio-temporal resolution. Two sets of tests are set up, focusing on the precipitation process on 2 February 2019, Test 1 is a 6h simulation (0:00 to 6:00 on 2 February) with 15 s and 60 s integration at the same grids (15km resolution) of this study. Test 2 is a 24-h nested simulation (from 00:00 on 2 February to 00:00 on 3 February) using 9 km (54 s) and 3 km (18 s) spatial (temporal) resolution, with the model domain shown in Fig. R1 (because the 15km:3km resolution 1:5 scale nested simulation cannot run under the model setting of this study, the 1:3 scale is used to ensure the stability of the calculation). The tests adopt the same model settings as the SBM-NEW experiment except that four-dimensional assimilation is not used.

[Figure]

**Figure R1.** The model domain of the test

Fig. R2 presents $N_d$, cloud liquid water content (CLWC), water vapor content and supersaturation at 925 hPa with active aerosol-cloud processes simulated at different temporal resolutions from Test 1. The SBM simulations at 60s and 15s resolutions show the overall similar distribution, with only minor differences in $N_d$ due to differences in resolving meteorological conditions and aerosol activation. The differences between the simulations of aerosol activation at the two temporal resolutions are specifically analyzed by selecting nine grid points according to $N_d$ from high to low (locations shown in Fig. R2a). The analysis shows (Fig. R3) that there are some differences in the simulated $N_d$ at different temporal resolutions, but the 60s resolution does not show clear negligence of activation processes and differences in the overall variation compared to the 15s resolution, and the differences in the simulated aerosol activation at the two resolutions mainly lie in the differences in meteorological and aerosol conditions at different time steps.

[Figure]

**Figure R2.** $N_d$ (in $cm^{-3}$, a and e), cloud liquid water content (CLWC, in $g \cdot m^{-3}$, b and f), water vapor content (in $g \cdot m^{-3}$, c and g) and supersaturation (in %, d and h) simulated with 60 s (a-d) and 15 s (e-h) integration and 15 km spatial resolution at 925 hPa.

[Figure]

**Figure R3.** $N_d$ (in $cm^{-3}$, black lines), supersaturation (in %, green lines), water vapor content (in $g \cdot m^{-3}$, blue lines) and CLWC (in $g \cdot m^{-3}$, red lines) simulated with 60s (light-colored lines) and 15s (deep-colored lines) integration (the locations of a-i are shown in Fig. R2a)

Test 2 then examines the ability of coarse spatio-temporal resolution to simulate aerosol-cloud variations. In test 2, only the data of d01 in the same area as d02 is selected to make the two match. As shown in Fig. R4, similar column AOD, $N_d$ and precipitation as well as 925 hPa (layer with

active aerosol-cloud processes) CLWC, $N_d$ and supersaturation distributions are simulated at 9 km and 3 km resolutions, while the 3 km grid relies on its finer resolution to simulate more detailed aerosol and cloud distribution information and stronger central intensity of precipitation. The analysis of the temporal variations of cloud parameters at each point on 925 hPa (sampled from high to low by $N_d$, locations shown in Fig. R4h) presented in Fig. R5 also supports the conclusion of Test 1 that the coarse resolution simulation differs only slightly from the fine resolution simulation in resolving specific processes due to differences in resolving meteorological fields, but both show consistent overall variations. In addition, the statistics on the model grid indicate that $N_d$ with aerosol (Fig. R6) and CLWC with $N_d$ (Fig. R7) show consistent trends at 9 km and 3 km resolutions, with the difference that the high-resolution simulation provides richer detail as well as a relatively larger number of intense activation samples and higher average $N_d$ due to more fine resolving of the meteorological field. The lack of detail is compensated by more samples at a relatively larger spatial scale and longer time scale, and the lack of accuracy in resolving the meteorological field can be compensated by (1) optimization of the meteorological field simulation by four-dimensional assimilation, (2) resolving of sub-grid convection by cumulus parameterization, (3) inclusion of more abundant atmospheric processes at a larger spatial and temporal scale, (4) selection of winter period dominated by cloud types with large temporal and spatial ranges such as stratocumulus, and (5) filtering out of ice-phase processes when analyzing.

Due to (1) the better simulation performance of SBM compared to the bulk scheme for cloud macroscopic and microscopic properties at 15 km resolution demonstrated in Sect. 3.2, (2) the consistency of the simulated aerosol and cloud variations at coarse resolution with fine resolution revealed by the tests, (3) the aforementioned measures to compensate for the lack of meteorological field simulation ability at coarse resolution, and (4) the convenience of SBM for resolving aerosol particle size, aerosol composition and droplet size on aerosol-cloud interaction, it is appropriate to use 15 km in this study.

[Figure]

**Figure R4.** d01 (a-c and g-i) and d02 (d-f and j-l) simulated column AOD (dimensionless), $N_d$ (in cm$^{-3}$) and precipitation (in mm) as well as 925 hPa CLWC (in g·m$^{-3}$), $N_d$ (in cm$^{-3}$) and supersaturation (in %)

[Figure]

**Figure R5.** d01 (light-colored lines) and d02 (deep-colored lines) simulated 925 hPa $N_d$ (in $cm^{-3}$, black lines), supersaturation (in %, green lines), water vapor content (in $g \cdot m^{-3}$, blue lines) and CLWC (in $g \cdot m^{-3}$, red lines) variations (a-f selected from highest to lowest according to $N_d$, as shown in Fig. R4h)

[Figure]

**Figure R6.** Probability density distribution functions and means (lines in the figures) of the simulated $N_d$ relative to aerosol number concentration ($N_{aero}$) and $PM_{10}$ (sum of probabilities corresponding to 1 for each $N_{aero}$ or $PM_{10}$ value) in d01 (a and c) and d02 (b and d)

[Figure]

**Figure R7.** Probability density distribution functions (sum of probabilities corresponding to 1 for each $N_d$ value) and means (lines in the figures) of CLWC relative to $N_d$ of precipitation clouds (a-b) and non-precipitation clouds (c-d) in d01 (a and c) and d02 (b and d)

---

## Author Comment (AC2)

**Response to the Comments of Referees**

**Journal:** Atmospheric Chemistry and Physics
**Manuscript Number:** egusphere-2023-331
**Title:** Exploring aerosol-cloud interactions over eastern China and its adjacent ocean using the WRF-SBM-MOSAIC model
**Author(s):** Jianqi Zhao, Xiaoyan Ma, Johannes Quaas, and Hailing Jia

We thank the reviewers and editor for providing helpful comments to improve the manuscript. We have revised the manuscript according to the comments and suggestions of the referees.

The referee's comments are reproduced (black) along with our replies (blue). All the authors have read the revised manuscript and agreed with the submission in its revised form.

**Anonymous Referee #2**

This paper investigated the interactions of aerosols and clouds over eastern China (EC) and its adjacent ocean region (ECO) during wintertime based on WRF-Chem with the SBM scheme by coupling the online aerosol module. The results show that the cloud variables are simulated more precisely compared to the bulk model and the default SBM scheme. Besides, the use of the four-dimensional data assimilation is evaluated using multiple observations and shows a positive effect on the simulation results. Upon all these improved models and methods, the authors analyze the differences in aerosol-cloud interactions over EC and ECO owing to the distinct aerosol physical and chemical properties and the meteorological conditions and examine the variations of cloud droplet number concentration with the increase of aerosol number concentration. Moreover, the rapid adjustments for precipitation clouds and non-precipitation clouds are discussed with the variations of cloud liquid water content and cloud effective radius over EC and ECO. This seems like tremendous work exploring aerosol-cloud interactions.

We thank the reviewer for taking the time to assess the manuscript. The reviewer's comments provide great help to improve our research and address deficiencies. We have revised the manuscript carefully according to the reviewer's comments. Please see the following detailed point-by-point responses.

**Major comments:**

1. One concern is about the evaluation of aerosol simulation. This study only evaluated the simulation of near-surface $PM_{2.5}$, which is not sufficient considering that the aerosol effects on clouds act at a certain altitude. An evaluation of aerosol vertical profile or at least aerosol optical depth should be added.

   Thanks for the suggestion. We have added the evaluation of simulated aerosol optical depth in the revised manuscript (Figure 4d-f and lines 298-302).

2. The authors selected the liquid-only clouds from MODIS data based on some criteria, and the cloud top is detected by MODIS. To compare with MODIS, how to pick the liquid-only clouds from the model output? How is the cloud top of model output defined to evaluate COT, CLWP, CER, and $N_d$?

   In our submitted manuscript, liquid-phase clouds are defined when the clouds with liquid cloud

water content and cloud fraction above 0. Considering that the differences in satellite retrievals and model parameterization calculations, many previous studies defined the liquid-phase clouds in the models based on certain thresholds when comparing with satellite-retrieved data, for example, Roh et al. (2020, https://doi.org/10.1175/JAS-D-19-0273.1) classified the clouds with CLWC > 1 mg $m^{-3}$ and cloud ice water content (CIWC) < 1 mg $m^{-3}$ as liquid-phase clouds. Therefore, we redefined liquid-phase cloud in the revised manuscript, based on the selection of column COT $\geq$ 5 that matched with MODIS filtering, the vertical layers (48 layers in total) with cloud optical thickness for water (COTW) > 0.1 and cloud optical thickness for ice (COTI) < 0.01 at each grid point and each time are selected as liquid-phase cloud layers, and the highest layer meeting this condition is the simulated cloud top (this filtering of simulated data is only used for comparison with MODIS data, and the analysis of aerosol-cloud interactions in liquid-phase clouds in this study is strictly limited to CLWC > 0 and CIWC = 0). We add this statement in lines 269-276.

3. For the precipitation clouds investigated in this study, the simulated precipitation should be compared with observational data.

   Thanks for the reminder. We have added the evaluation of simulated precipitation in the revised manuscript (Figure 4a-c and lines 296-298).

4. Do the samples in Figure 11-12 contain only liquid water? Could the samples contain the liquid part of the mixed clouds? How to exclude the influence of other types of clouds considering the mechanism of cloud formation and development varies with the cloud type?

   We are grateful for this suggestion, it is a point we had not considered before. In the revised manuscript, we have imposed a strict restriction on cloud phase, selecting only the grids with CLWC > 0 and CIWC = 0 (CIWC is the sum of ice, snow, graupel and hail water contents) as liquid-phase clouds, and clarified this restriction in lines 275-276. In addition, we clarified in the manuscript title and content that this study targets liquid-phase cloud.

5. As for the $N_d$ variations in each $N_{aero}$ interval in Figure 11, at the second stage, the authors stated that $N_d$ in EC still increases swiftly with $N_{aero}$ due to the relatively strong updraft and surface radiative cooling. Please explain why the $N_d$ in ECO does not increase like $N_d$ in EC.

   In ECO, due to the inability to rely on the effects of surface like EC to reach supersaturation, the dominance of water vapor variation on aerosol activation is more pronounced, and the supersaturation shows a steady decreasing trend with increasing $N_d$. After $N_{aero}$ exceeds 10000 cm$^{-3}$ (average $N_d$ exceeds 500 cm$^{-3}$), the increase in small aerosols and the decrease in supersaturation prevents its $N_d$ from continuing to increase and $N_d$ starts to show a decreasing trend. We add this explanation in lines 458-461 and add a detailed analysis of the difference between EC and ECO atmospheric supersaturation pathway in lines 410-429.

6. Many studies using satellite data to explore aerosol-cloud interactions view AOD or aerosol index (AI) as an indicator of aerosol concentration due to the limit of observations. I wonder if the simulated AOD is improved after the assimilation. It would be great if the authors could plot the variations of the simulated CLWP with AOD in EC and ECO.

   Thanks for the suggestion. We have added the evaluation of the effect of assimilation on AOD simulation in the revised manuscript (Figure 4d-f and lines 298-302). We have also plotted the variations of the simulated CLWP with AOD, and added an analysis of the variations of CLWP and

its relevant influences with AOD in Figure 15 and lines 522-537.

7. In section 3.4, when exploring aerosol-cloud interactions, meteorological fields may affect both aerosols and clouds, resulting in covariance between the two, such that changes in clouds cannot be attributed to aerosols. The authors need to further discuss the role of the meteorological field in this study and exclude its effects.

Thanks for the suggestion. We have discussed the variations of $N_d$ and CLWC under different meteorological (U-wind, V-wind, W-wind, temperature, water vapor content, temperature variation and water vapor variation) and aerosol conditions in the revised manuscript (Fig. 13 and Fig.16 as well as lines 475-491 and lines 546-551).

**Minor comments:**

1. Line 9: Delete "of". "Coupling a spectral-bin cloud…" is suitable.

Corrected .

2. Line 22: Delete ", which".

Corrected.

3. Line 22: It should be "large-scale"

Corrected.

4. Line 28: Aerosols show indirect effects as CCN and IN.

Corrected.

5. Line 29: "Remain" should be changed to "remains".

Corrected.

6. Line 51: It should be "depends on" instead of "depends".

Corrected.

7. Line 86: It should be "Benefiting from advances in computational science".

Corrected.

8. Line 115: It should be "is" before "consistent".

Corrected.

9. Line 130: "Thus greatly promoting" is more suitable. So is "optimizing" in Line 270.

Corrected.

10. Line 260: Replace "This" with "These".

Corrected.

11. Line 355: Replace "comes" with "come" and delete "are" at the end of this line.

Corrected.

12. Line 370: Is there any direct evidence to prove that radiative cooling makes a large number of cloud droplets distributed near the surface?

Since surface radiative cooling occurs mainly from night to early morning, we analyze the variations of temperature profile and near-surface supersaturation in EC during daytime (7:00 to 18:00 in Beijing time) and nighttime (19:00 to 06:00 of the next day in Beijing time), and the results are shown in Fig. S2b and c. It can be seen that the presence of nighttime near-surface thermal inversion makes the nighttime supersaturation generally higher than the daytime during the simulation period, which effectively boosts the aerosol activation and indicates the important influence of surface longwave radiative cooling on atmospheric supersaturation and aerosol activation. We add this analysis to lines 419-422 of the revised manuscript.

13. Figure 11 shows the variation of $N_d$ with aerosol and other related factors based on the statistics of the model grids with CF greater than 0 at each time. The method to calculate CF in this study is that CF equals 1 when the sum of cloud water and cloud ice mixing ratios is greater than $10^{-6}$ kg·kg$^{-1}$, otherwise, the CF equals 0. Why does the calculation of CF use cloud ice mixing ratio since this study focuses on liquid clouds?

Thanks for the comment. The CF in the original manuscript is calculated from the model parameterization (the threshold method mentioned in this comment). In the original manuscript we sampled the grids with CF and CLWC greater than 0. In the revised manuscript, we have imposed a strict restriction to exclude the ice phase, i.e. only the grids with CLWC>0 and CIWC=0 are selected.

14. Figure 6-8: Have you done a significance test for the correlation coefficient?

We calculated spatial correlation based on Pearson product-moment method, which is not accompanied by a significance test. Due to the spatial and temporal discontinuity of MODIS data and our data filtering, only some of spatial coordinates have relatively continuous time series, and many coordinates have only one or a few valid values, which makes it meaningless to do significance tests for each coordinate point, so we did not perform significance test for the spatial correlation. In addition, we performed significance tests on the $PM_{2.5}$ data with good continuity, and we used RMSE to auxiliary spatial correlation coefficients.

15. How do the authors define the liquid cloud? I just wonder why the liquid clouds appear higher than 4 km in Figure 10b. Besides, should the role of sea salt acting as the ice nuclei be considered in the discussion?

In the submitted manuscript, we did not strictly select the grids containing only liquid-phase clouds for the analysis of cloud droplet distributions in Fig. 10. Because aerosol activation can occur at any place where supersaturation and aerosol conditions are met, cloud droplets can appear above 4 km. In the revised manuscript, we only analyze the cloud droplets in the liquid-phase clouds, and the new $N_d$ distribution can be seen in Fig. 11c-d, where almost no liquid-phase clouds appear above 4 km. The coupling of SBM and MOSAIC by us and most related researchers focus mainly on the activation of aerosols into cloud droplets without modifications to the SBM ice phase nucleation, so the role of sea salt acting as ice nuclei cannot be directly resolved. In addition, we make a further statement in the title and content of the article to explore only aerosol-cloud interactions in liquid-phase clouds, so the role of sea salt acting as an ice nuclei is not discussed here.

16. What do the downdrafts in Figure 10d imply?

Thanks for the comment. The figure presents the spatial distribution of the atmospheric vertical velocity, where both updrafts and downdrafts appear over the whole spatial scale.

17. The manuscript mentioned that the near-surface areas around 29°N and 31°N (Fig. 10b) exhibit high atmospheric supersaturation due to the effect of topographic uplift (Fig. 10c). Why the cloud number concentration is rather low in the south of the topographic uplift?

Thanks for the reminder. This is because Fig. 10 in our previous manuscript was based on oblique profiles (along the yellow lines in Fig. R1a), which did not provide enough complete information. We modify it by using the average value of the corresponding latitude instead of the oblique profile to make the figure present more reasonable information.

[Figure]

**Figure R1**. Topography (unit: m) of the model domain, MICAPS (a) and assimilated simulated (b) 850 hPa wind fields (unit: m·s-1) and positions of the oblique lines (yellow lines in Fig. R1a)